# Hypoxia-induced proteasomal degradation of DBC1 by SIAH2 in breast cancer progression

Qiangqiang Liu[1], Qian Luo[1], Jianyu Feng[1], Yanping Zhao[2], Biao Ma[1], Hongcheng Cheng[1], Tian Zhao[1], Hong Lei[1], Chenglong Mu[1], Linbo Chen[1], Yuanyuan Meng[1], Jiaojiao Zhang[1], Yijia Long[1], Jingyi Su[1], Guo Chen[1], Yanjun Li[1], Gang Hu[2], Xudong Liao[1], Quan Chen[1], Yushan Zhu[1]*

[1]State Key Laboratory of Medicinal Chemical Biology, Frontiers Science Center for Cell Responses, Tianjin Key Laboratory of Protein Science, College of Life Sciences, Haihe Laboratory of Cell Ecosystem, Nankai University, Tianjin, China; [2]School of Statistics and Data Science, LPMC and KLMDASR, Nankai University, Tianjin, China

**Abstract** DBC1 has been characterized as a key regulator of physiological and pathophysiological activities, such as DNA damage, senescence, and tumorigenesis. However, the mechanism by which the functional stability of DBC1 is regulated has yet to be elucidated. Here, we report that the ubiquitination-mediated degradation of DBC1 is regulated by the E3 ubiquitin ligase SIAH2 and deubiquitinase OTUD5 under hypoxic stress. Mechanistically, hypoxia promoted DBC1 to interact with SIAH2 but not OTUD5, resulting in the ubiquitination and subsequent degradation of DBC1 through the ubiquitin–proteasome pathway. *SIAH2* knockout inhibited tumor cell proliferation and migration, which could be rescued by double knockout of *SIAH2/CCAR2*. Human tissue microarray analysis further revealed that the SIAH2/DBC1 axis was responsible for tumor progression under hypoxic stress. These findings define a key role of the hypoxia-mediated SIAH2-DBC1 pathway in the progression of human breast cancer and provide novel insights into the metastatic mechanism of breast cancer.

*For correspondence:
zhuys@nankai.edu.cn

Competing interest: The authors declare that no competing interests exist.

## Editor's evaluation

This is a study with relatively convincing data on examining the dynamic regulation of hypoxia regulating SIAH2/OTUD5 interaction with DBC1 by regulating DBC1 stability. Therefore, DBC1, by regulating p53 signaling, would affect breast cancer phenotype in vivo.

## Introduction

The occurrence and development of tumors are modulated by the dual regulation of genetic instability and the tumor microenvironment (*Singleton et al., 2021*). Hypoxic stress or low oxygen tension, a major hallmark of the tumor microenvironment, plays an essential role in the progression and metastasis of many solid tumors (*Cheng et al., 2020*; *Lee et al., 2019*). Moreover, our previous studies have documented that hypoxic stress attenuates tumorigenesis and progression by modulating the Hippo signaling pathway and mitochondrial biogenesis (*Ma et al., 2015*; *Ma et al., 2019*). To extensively understand the critical roles of hypoxic stress in tumorigenesis or tumor development, more precise mechanisms need to be further explored.

Deleted in breast cancer 1 (DBC1; also known as CCAR2) is a nuclear protein containing multifunctional domains and plays a critical role in a variety of cancers. Importantly, DBC1 cooperates with lots

of epigenetic and transcriptional factors to regulate cell activities. DBC1 mediates p53 function not only by inhibiting the deacetylase activity of SIRT1, but also by inhibition of p53 ubiquitination and degradation (*Qin et al., 2015*; *Akande et al., 2019*; *Kim et al., 2008*; *Zhao et al., 2008*). In addition, DBC1 can specifically inhibit deacetylase HDAC3 activity and alter its subcellular distribution, resulting in cell senescence (*Chini et al., 2010*). Most importantly, mice with a genetic deletion of *CCAR2* exhibited more likely to spontaneously develop lung tumors, liver tumors, lymphomas, and teratomas and showed poor overall survival (*Qin et al., 2015*). Recently, the downregulation of DBC1 highly correlates with poor prognosis and distant metastatic relapse in breast, colon, and prostate cancer patients, and low levels of DBC1 determine tumor grade and metastasis (*Won et al., 2015*; *Noguchi et al., 2014*; *Yu et al., 2013*). Accumulating evidence has shown that DBC1 activity and quality of control play key roles in tumorigenicity, while the mechanisms by which DBC1 stability is regulated remain elusive.

Protein ubiquitination is one of the most-studied post-translational modifications and is critical and essential for protein stability, activity, localization, and biological function. Here, we delineated that the ubiquitination and stability of DBC1 were orchestrated by the E3 ubiquitin ligase SIAH2 and deubiquitinase OTUD5 under normoxic or hypoxic stress. Mechanistically, hypoxic stress promoted the interaction between DBC1 and SIAH2 and enhanced the disassociation of DBC1 from OTUD5, resulting in an increase in DBC1 ubiquitination and degradation, contributing to tumor cell proliferation and migration. Human tissue microarray analysis further revealed that the SIAH2/DBC1 axis was responsible for regulating tumor progression under hypoxic stress. Our findings provide novel insights into the metastatic mechanism of breast cancer and a promising therapeutic target for breast cancer.

## Results

### Hypoxic stress triggers the degradation of DBC1

Hypoxia is a common hallmark of solid tumors and contributes to the development and progression of many cancers (*Lee et al., 2019*). To investigate the effects of hypoxia on breast cancer cells, we first performed RNA-seq analysis of MDA-MB-231 cells in response to hypoxic stress. Differential expression analysis showed that 1151 genes were significantly upregulated and 310 genes were downregulated (adjusted p<0.05) under hypoxic stress (*Figure 1A*). Enrichment analysis of differentially expressed genes by Metascape suggested that the upregulated genes were related to the pathways of cell proliferation and cancer; in contrast, the downregulated genes were implicated in the DNA damage repair, senescence, SIRT1, and p53 signaling pathways (*Figure 1B*). To further investigate the mechanism by which hypoxia regulates the p53 signaling pathway, we examined p53 pathway activity by Western blotting and found that the stabilities of SIRT1 and p53 protein were not changed under hypoxia, but the acetylation of p53 was decreased (*Figure 1C*). Interestingly, we observed that the protein level of DBC1 gradually decreased with the duration of hypoxia (*Figure 1D*). In order to verify whether DBC1 protein level is involved in the HIF1α signaling pathway, we further analyzed DBC1 protein levels with HIF1α inhibitor under hypoxic conditions. The results showed that hypoxia-induced decrease in DBC1 protein level and p53 signaling pathway activity was not blocked by HIF1α inhibitor (*Figure 1—figure supplement 1A*). These results prove that HIF1α signaling pathway is not involved in hypoxia-induced decrease in DBC1. Under hypoxic conditions, the decrease in DBC1 protein level could be inhibited by the proteasome inhibitor MG132 (*Figure 1E*), but not the lysosomal inhibitor BA1 (*Figure 1—figure supplement 1B and C*), suggesting that hypoxia induced DBC1 degradation through the ubiquitin–proteasome system. In most cell lines derived from different subtypes of breast cancer or other cancers, DBC1 protein was degraded under hypoxia, suggesting the reduced stability of DBC1 was a general rule in hypoxic conditions (*Figure 1F* and *Supplementary file 1*), while the mRNA level of *CCAR2* was not changed under hypoxia (*Figure 1G* and *Supplementary file 1*). Similarly, the RNA-seq datasets of *CCAR2* knockout MDA-MB-231 cells showed that genes related to cell proliferation and migration were also upregulated, and genes associated with DNA repair and apoptosis were downregulated (*Figure 1H and I*, *Figure 1—figure supplement 1D and E*). Collectively, these results suggest that hypoxia and DBC1 degradation contribute to tumor progression, which suggests that hypoxia may regulate tumor progression by mediating DBC1 protein degradation.

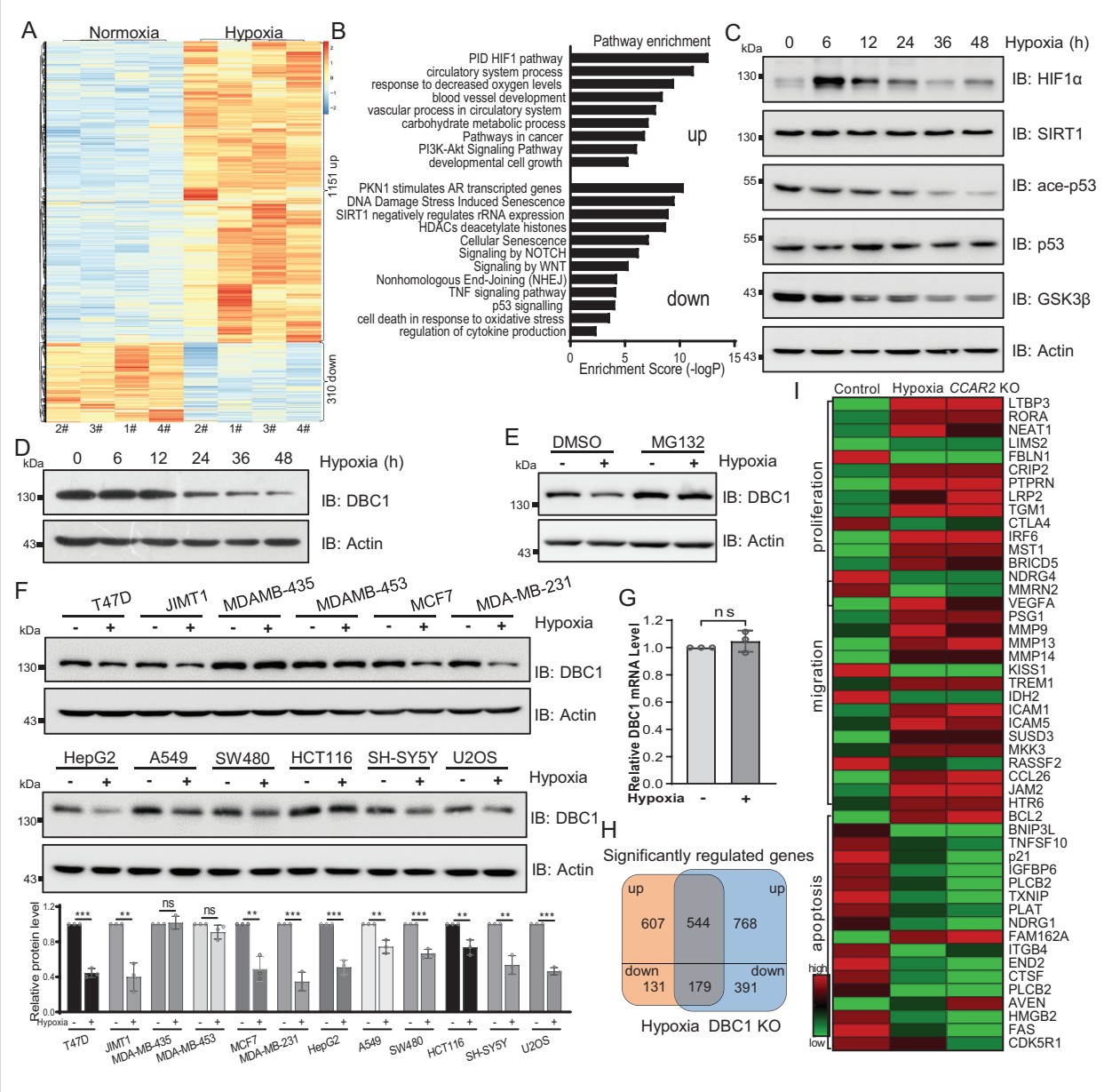

**Figure 1.** Hypoxia induces DBC1 degradation. (**A**) MDA-MB-231 cells were cultured under normoxia or hypoxia for 24 hr and then the total RNA was extracted from cells using a total RNA extraction kit. RNA-sequence analysis of gene expression and heatmap shows the differential transcriptomic expression. (**B**) Highlights of the enriched pathways from transcriptomic expression by Metascape. (**C**) MDA-MB-231 cells were exposed to hypoxia for the indicated time and then Western blotting analysis of the protein levels using the indicated antibodies. (**D**) MDA-MB-231 cells were exposed to hypoxia for the indicated time and then protein levels of DBC1 were detected by Western blotting. (**E**) MDA-MB-231 cells were exposed to normoxia or hypoxia for 24 hr, with or without MG132 (10 μM), and then the protein levels of DBC1 were detected by Western blotting. (**F**) The indicated cell lines were cultured under normoxia or hypoxia for 24 hr and then the protein level of DBC1 was detected by Western blotting. Mean ± SEM from three independent experiments, Student's t-test, ns, not significant, **p<0.01, ***p<0.001. (**G**) MDA-MB-231 cells were cultured under normoxia or hypoxia for 24 hr. Cells were collected and lysed with RNA lysis buffer. The mRNA level of DBC1 was quantified by quantitative real-time PCR assay (data are the mean ± SEM of three experiments, Student's t-test, ns, not significant). (**H**) Venn diagram showing the numbers of DBC1-regulated genes common to hypoxia target genes. (**I**) Heatmap showing the representatives of differentially regulated transcripts associated with tumor initiation and progression.

The online version of this article includes the following source data and figure supplement(s) for figure 1:

**Source data 1.** Hypoxia induces DBC1 degradation.

**Figure supplement 1.** The stability of DBC1 was regulated under hypoxic conditions.

**Figure supplement 1—source data 1.** Hypoxia induces DBC1 degradation.

## SIAH2 interacts with DBC1 and regulates its stability

To identify the E3 ligases that potentially ubiquitinates DBC1 to contribute to its degradation, several ubiquitin E3 ligases previously reported to be involved in the hypoxia response were screened (*Figure 2A*, *Figure 2—figure supplement 1A*), and SIAH2 was finally identified to be responsible for the stability of DBC1 (*Figure 2B*). To confirm our hypothesis, LC-MS/MS analysis of proteins bound to SIAH2$^{RM}$ (an SIAH2 enzyme-inactive mutant) was performed. DBC1 was validated as one of the substrates of SIAH2 (*Supplementary file 1*). To further explore the relationship between SIAH2 and DBC1, we carried out a co-immunoprecipitation (Co-IP) assay and found that exogenously expressed Flag-SIAH2$^{RM}$ and Myc-DBC1 were reciprocally co-immunoprecipitated (*Figure 2C*). Meanwhile, we validated the interaction between endogenous SIAH2 and DBC1 (*Figure 2D*, *Figure 2—figure supplement 1B and C*), and purified glutathione S-transferase (GST) SIAH2 was able to pull down His-DBC1 in vitro (*Figure 2E*), suggesting that DBC1 could directly interact with SIAH2. To identify the necessary domain of DBC1 responsible for interaction with SIAH2, we constructed several truncations of DBC1 and SIAH2 (*Figure 2—figure supplement 1D and E*), and the Co-IP assay showed that the 1–230 amino acids at the N-terminus of DBC1 were necessary for binding to SIAH2 (*Figure 2F*), and the full-length SIAH2 was required for their interaction (*Figure 2G*). Importantly, the N-terminus (1–461) of the DBC1 protein rather than the C-terminus (462–923) was pulled down by SIAH2 in vitro (*Figure 2H and I*). These results suggest that the N-terminus of DBC1 is crucial for its interaction with SIAH2. In line with the fact that SIAH2 acts as an E3 ligase, mediating substrate degradation by the ubiquitin–proteasome pathway, we found that SIAH2, rather than SIAH2$^{RM}$, reduced the protein level of endogenous DBC1 in a dose-dependent manner (*Figure 2J and M*) and had no effect on the transcriptional level of DBC1 (*Figure 2—figure supplement 1F* and *Supplementary file 1*). The CHX chase assay showed that ectopic expression of wildtype SIAH2, but not SIAH2$^{RM}$, promoted DBC1 degradation over time (*Figure 2K and L* and *Supplementary file 1*). Furthermore, the proteasome inhibitor MG132 completely reversed DBC1 destabilization (*Figure 2N*), whereas the lysosomal inhibitor bafilomycin A1 (BA1) had no such function (*Figure 2—figure supplement 1G*), suggesting that SIAH2 mediated DBC1 degradation through the proteasome pathway rather than lysosomes. These results reveal that SIAH2 promotes DBC1 degradation by directly interacting with DBC1.

## SIAH2 is responsible for DBC1 ubiquitination and degradation under hypoxic stress

Next, we checked whether SIAH2 ubiquitinated DBC1, and the results showed that ectopic expression of SIAH2, but not SIAH2$^{RM}$, dramatically increased the ubiquitination level of DBC1 (*Figure 3A and B*). Additionally, immunoprecipitation analysis proved that SIAH2 induced the polyubiquitination of DBC1 in the form of K48 conjugation (*Figure 3—figure supplement 1A and B*). In vitro ubiquitination assay further revealed that DBC1 was directly ubiquitinated by SIAH2 rather than the E3 ligase-dead mutant SIAH2$^{RM}$ (*Figure 3C*). To identify the sites of DBC1 ubiquitinated by SIAH2, we performed MS analysis, and the results showed that K287 of DBC1 was the potential site ubiquitinated by SIAH2 (*Figure 3D*). Next, we found that ectopic expression of SIAH2 did not increase the ubiquitination level of the K287R mutant of DBC1, and the mutant did not degrade (*Figure 3E*, *Figure 3—figure supplement 1C and D*). In conclusion, these results demonstrate that SIAH2 ubiquitinates DBC1 to mediate its degradation. To further understand the mechanism underlying hypoxia-induced DBC1 degradation, we examined whether SIAH2 was responsible for DBC1 ubiquitination and degradation in response to hypoxia. Our results showed that under hypoxic stress both *SIAH2* deletion and K287R mutation of DBC1 failed to increase DBC1 ubiquitination (*Figure 3F and G*), and DBC1 degradation was also inhibited (*Figure 3H–J*, *Figure 3—figure supplement 1E and F*, and *Supplementary file 1*). Consistently, knockout of *SIAH2* prolonged the half-life of DBC1 under hypoxia (*Figure 3K and L*, *Figure 3—figure supplement 1F–H*, and *Supplementary file 1*). It has been reported that under hypoxic conditions the regulation of SIAH2 was regulated by several mechanisms including induction of gene expression, interference translation, and formation of dimeric complexes. SIAH2 could be phosphorylated by kinases such as p38 and Src under hypoxic conditions, increasing its ubiquitin ligase activity and protein stability (*Khurana et al., 2006*). When cells were treated with doramapimod, an inhibitor of p38, under hypoxic conditions, we found that the degradation of DBC1 was partially blocked (*Figure 3—figure supplement 1I*). Moreover, we found that inhibition of p38 partially decreased SIAH2 protein level and inhibited the interaction between SIAH2 and DBC1 under

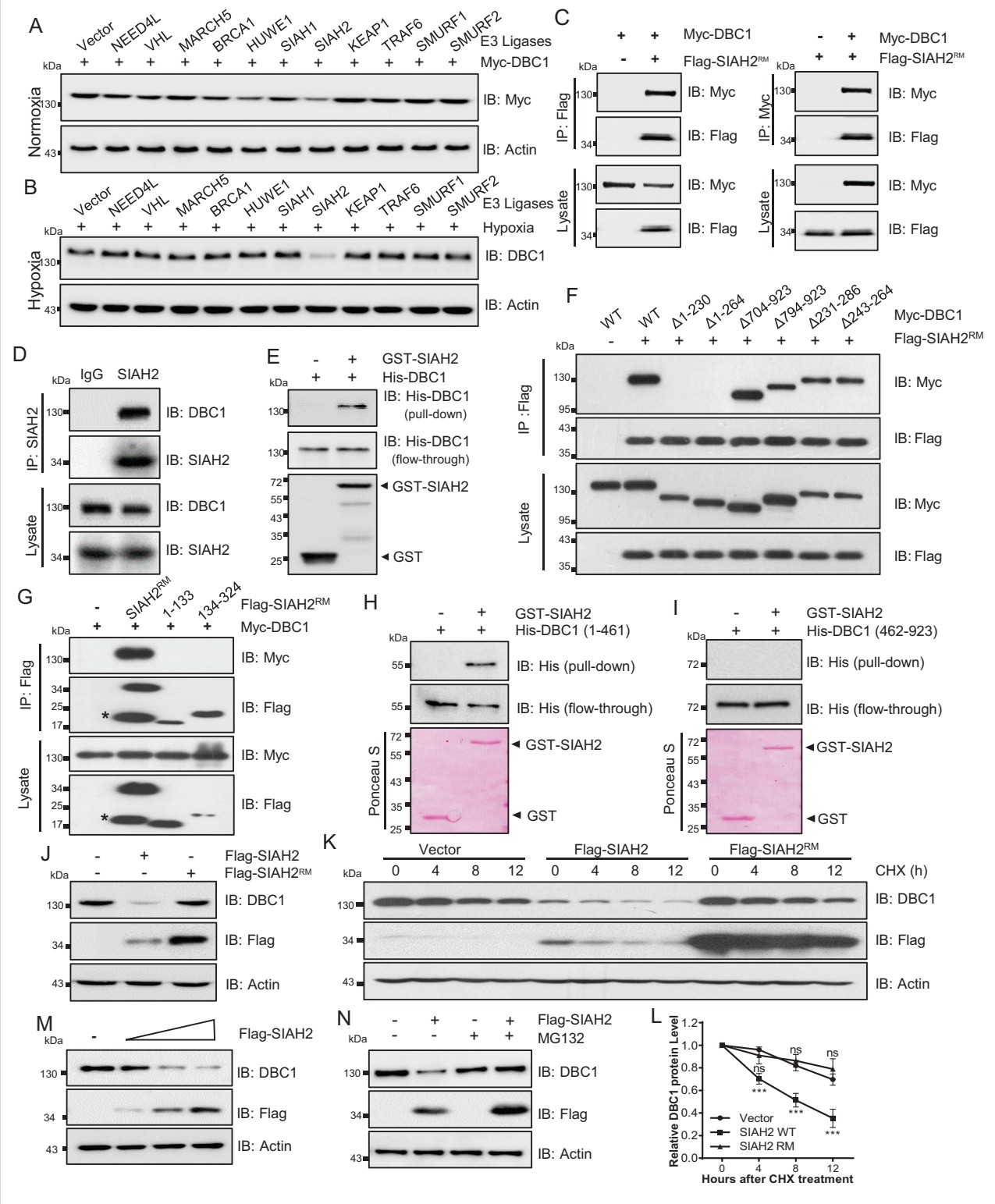

**Figure 2.** SIAH2 interacts with DBC1 and regulates its stability. (**A**) HeLa cells were transfected with Myc-DBC1 and several hypoxia-relative E3 ligases and cultured under normoxic conditions for 24 hr. The protein level of DBC1 was detected by Western blotting. (**B**) HeLa cells were transfected with several hypoxia-relative E3 ligases and cultured under hypoxic conditions for 24 hr. The protein level of DBC1 was detected by Western blotting. (**C**) HEK293T cells were transfected with Myc-DBC1 and Flag-SIAH2^RM for 24 hr. Cells were collected for immunoprecipitation with anti-Flag or anti-Myc antibody. (**D**) MDA-MB-231 cells were cultured under hypoxia for 18 hr, then treated with 10 μM MG132 and incubated under normoxia or

*Figure 2 continued on next page*

*Figure 2 continued*

hypoxia for another 6 hr. Endogenous interactions between DBC1 and SIAH2 were analyzed by co-immunoprecipitation (Co-IP). (**E**) Purified GST and GST-tagged SIAH2 proteins were used for GST affinity isolation of His-DBC1 in vitro and blotted with an anti-DBC1 antibody. (**F**) HEK293T cells were co-transfected with full-length or truncated forms of Myc-DBC1 and Flag-SIAH2^RM, and immunoprecipitation was performed with an anti-Flag antibody. Co-immunoprecipitated DBC1 and SIAH2 were detected by Western blotting with anti-Myc and anti-Flag antibodies, respectively. (**G**) Myc-DBC1 was co-transfected with full-length or truncated forms of Flag-SIAH2^RM, and immunoprecipitation was performed with an anti-Myc antibody. Co-immunoprecipitated SIAH2 and DBC1 were detected by Western blotting with anti-Flag and anti-Myc antibodies, respectively. (**H, I**) GST pull-down analysis showing the direct interaction between bacterially expressed GST-SIAH2 and the N- or C-terminal half of His-DBC1 in vitro. (**J**) HeLa cells were transfected with Flag-SIAH2, Flag-SIAH2^RM, or the empty Flag-vector for 24 hr and the cell lysates were subjected to Western blotting analysis of DBC1. (**K**) HeLa cells were transfected with Flag-SIAH2, Flag-SIAH2^RM, or the empty Flag-vector for 24 hr. CHX (10 μM) was added for the indicated time, and the cell lysates were subjected to Western blotting analysis of DBC1. (**L**) Quantification of DBC1 protein level in (**K**) (mean ± SEM from three independent experiments, two-way ANOVA, ns, not significant, ***p<0.001). (**M**) HeLa cells were transfected with Flag-SIAH2 by quantitative gradient and then protein level of DBC1 was detected by Western blotting. (**N**) HeLa cells were transfected with Flag-SIAH2 for 24 hr and then treated with or without MG132 (10 μM) for 6 hr. The cell lysates were subjected to Western blotting analysis of DBC1.

The online version of this article includes the following source data and figure supplement(s) for figure 2:

**Source data 1.** SIAH2 interacts with DBC1.

**Figure supplement 1.** The stability of DBC1 was regulated by SIAH2.

**Figure supplement 1—source data 1.** SIAH2 interacts with DBC1.

hypoxia (*Figure 3—figure supplement 1J*). Therefore, hypoxia activates SIAH2 to interact with DBC1, which induces DBC1 degradation by ubiquitin–proteasome system to regulate DBC1 stability. Under hypoxic conditions, SIAH2 acts as an E3 ligase-related hypoxia to target substrate degradation. Taken together, these results demonstrate that hypoxia-induced DBC1 degradation is dependent on SIAH2-mediated DBC1 ubiquitination.

## OTUD5 regulates the deubiquitination and stability of DBC1

Strikingly, we also observed that the ubiquitination level of DBC1 was sharply decreased when returned to normal conditions after hypoxic stress (*Figure 4A*). Therefore, we screened the deubiquitinating enzymes (DUBs) plasmids library to identify the candidate responsible for deubiquitination of DBC1 (*Figure 4—figure supplement 1A*). Lots of interaction assays suggested that the potential DUB OTUD5 might interact with DBC1 (*Figure 4—figure supplement 1A and B*) and regulate its ubiquitination (*Figure 4—figure supplement 1C*). Co-IP analysis demonstrated the exogenous interaction between OTUD5 and DBC1 (*Figure 4—figure supplement 1D and E*). Furthermore, we confirmed that endogenous DBC1 could directly interact with OTUD5 (*Figure 4B*, *Figure 4—figure supplement 1F*), which was further verified by an in vitro pull-down assay using purified His-tagged DBC1 to pull down OTUD5 from cell lysates (*Figure 4C*, *Figure 4—figure supplement 1G*). We next tested whether DBC1 deubiquitination was dependent on OTUD5 enzyme activity. The ubiquitination assay showed that only ectopic expression of wildtype OTUD5 but not inactive mutant C224S led to DBC1 ubiquitination level decrease under hypoxia (*Figure 4D*). Consistently, hypoxia-induced endogenous DBC1 degradation was specifically inhibited by expression of wildtype OTUD5 rather than the inactive mutant (*Figure 4E*). Moreover, the CHX assay also confirmed that ectopic expression of wildtype OTUD5 obviously prolonged the half-life of DBC1 (*Figure 4F and G* and *Supplementary file 1*), indicating that deubiquitination of DBC1 was beneficial for its stability. Domain mapping analysis further revealed that the N-terminal region (amino acid residues 1–230) of DBC1, which binds with SIAH2, was also necessary for its interaction with OTUD5 (*Figure 4H*). Interestingly, we found that either hypoxia treatment or overexpression of SIAH2 promoted the interaction between DBC1 and SIAH2, but inhibited the interaction of DBC1 with OTUD5 (*Figure 4I*, *Figure 4—figure supplement 1H*). Overall, our results suggest that SIAH2 and OTUD5 substitutionally interact with DBC1 in response to changes in oxygen concentration (*Figure 4J*), cooperatively regulating reversible DBC1 ubiquitination and stability to orchestrate DBC1 function.

## Hypoxia regulates tumor progression via the SIAH2-DBC1 axis

As suggested in *Figure 1*, hypoxia-induced DBC1 degradation might regulate pathways associated with tumor cell growth, such as SIRT1 and p53 signaling pathways. Therefore, we examined whether SIAH2-mediated DBC1 ubiquitination was responsible for tumor progression. Our results showed that

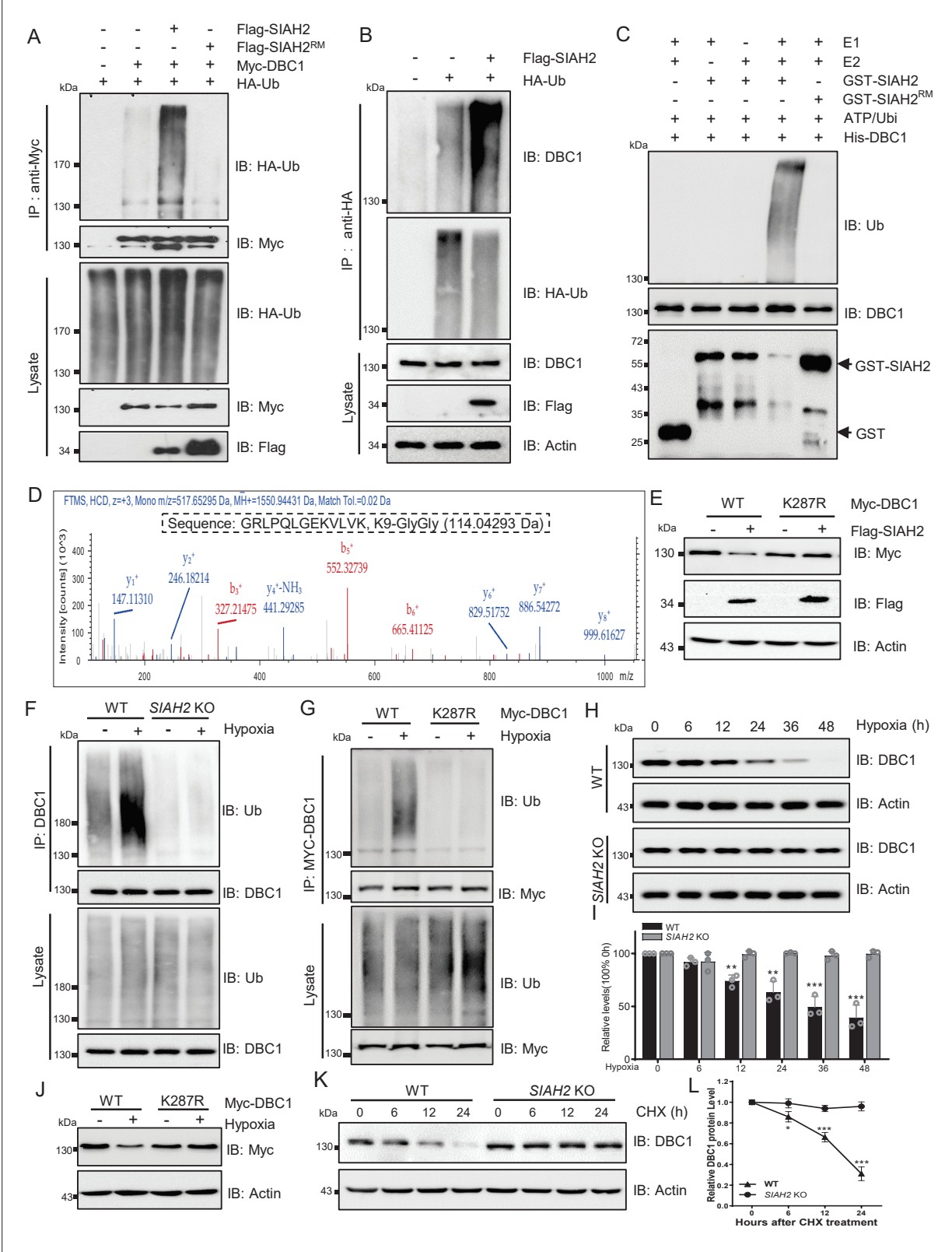

**Figure 3.** SIAH2 promotes DBC1 polyubiquitination and degradation under hypoxia. (**A**) HEK293T cells were transfected with Myc-DBC1, HA-Ub, and Flag-SIAH2 or Flag-SIAH2$^{RM}$ for 24 hr and then treated with MG132 (10 μM) for 6 hr. Ubiquitylation assays were performed and the ubiquitylation level of DBC1 was detected using an anti-HA antibody. (**B**) The HEK293T cells with or without Flag-SIAH2 transfection were treated with MG132 (10 μM) for 6 hr; then the cell lysates were subjected to anti-HA immunoprecipitation, and the immunoprecipitated were analyzed by Western blot using anti-DBC1.

*Figure 3 continued on next page*

*Figure 3 continued*

(**C**) Ubiquitination of bacterially expressed His-DBC1 by purified SIAH2 but not by SIAH2[RM] in vitro. The reactions were performed either with purified ubiquitin, UBA1 (E1), UBCH7 (E2), and SIAH2 or its inactive mutant or in the absence of UBA1, UBCH7, or ubiquitin. (**D**) Mass spectrometry analysis identifies K287 of DBC1 as the site for SIAH2-induced ubiquitination. Ubiquitinated DBC1 was trypsin digested and analyzed by LC-MS/MS. The MS/MS mass spectrum of a ubiquitinated peptide is shown for peptide DBC1 279–291 containing ubiquitinated Lys-287. (**E**) HeLa cells were transfected with Myc-DBC1 or the Myc-DBC1 (K287R) mutant and Flag-SIAH2 or the empty Flag-vector for 24 hr. The protein level of DBC1 was detected using an anti-Myc antibody. (**F**) WT and *SIAH2* knockout MDA-MB-231 cells were cultured under normoxia or hypoxia for 24 hr and then treated with MG132 (10 μM) for 6 hr. Cells were harvested, denatured, and lysed for immunoprecipitation with anti-DBC1 antibody. The ubiquitination level of DBC1 was assessed by immunoblotting with an anti-Ub antibody. (**G**) HeLa cells were transfected with Myc-DBC1 or the Myc-DBC1 (K287R) mutant and cultured under normoxia or hypoxia for 24 hr. Cells were harvested, denatured, and lysed for immunoprecipitation with anti-Myc antibody. The ubiquitination level of DBC1 was assessed by immunoblotting with an anti-Ub antibody. (**H**) WT and *SIAH2* knockout MDA-MB-231 cells were exposed to hypoxia for the indicated time and then protein level of DBC1 was detected by Western blotting. (**I**) Quantification of DBC1 protein as indicated in (**H**) (mean ± SEM from three independent experiments, Student's *t*-test, **p<0.01, ***p<0.001). (**J**) HeLa cells were transfected with Myc-DBC1 or the Myc-DBC1 (K287R) mutant and cultured under normoxia or hypoxia for 24 hr and then protein level of DBC1 was detected by Western blotting. (**K**) WT and *SIAH2* knockout MDA-MB-231 cells was exposed to hypoxia for 24 hr, CHX (10 μM) was added for the indicated time, and the cell lysates were subjected to Western blotting analysis of DBC1. (**L**) Quantification of DBC1 protein levels in (**K**) (mean ± SEM from three independent experiments, Student's *t*-test, *p<0.05, **p<0.01).

The online version of this article includes the following source data and figure supplement(s) for figure 3:

**Source data 1.** SIAH2 promotes DBC1 polyubiquitination and degradation.

**Figure supplement 1.** Hypoxia-induced DBC1 degradation is dependent on E3 Ligase SIAH2.

**Figure supplement 1—source data 1.** SIAH2 regulate DBC1 polyubiquitination and its degradation.

knockdown of *SIAH2*, in different breast cancer cell lines, attenuated the reduction of p53 pathway activity under hypoxia (***Figure 5A***). In addition, deletion of SIAH2 promoted etoposide triggering cell apoptosis under hypoxia, which could be rescued by simultaneous knockout of *CCAR2* (***Figure 5B***, ***Figure 5—figure supplement 1A***). The clone formation assay further revealed that cell growth was suppressed when SIAH2 was deleted under hypoxia, whereas double knockout of *SIAH2* and *CCAR2* almost did not inhibit cell growth (***Figure 5C***, ***Figure 5—figure supplement 1B***). Transwell and scratch assays also revealed that knockout of *SIAH2* inhibited cell migration under hypoxia rather than double knockout of *SIAH2* and *CCAR2* (***Figure 5D and E***, ***Figure 5—figure supplement 1C and D***). These results suggest that SIAH2 promotes cell growth, proliferation, and migration in response to hypoxia by ubiquitinating DBC1 to induce DBC1 degradation.

To validate the effects of SIAH2-mediated DBC1 stability on tumorigenesis and tumor progression in vivo, we implanted *SIAH2* knockout and *SIAH2/CCAR2* double-knockout MDA-MB-231 cells into the mammary fat pads of BALB/c nude mice and then monitored tumor growth. The results showed that both tumor volume and tumor weight in the nude mice transplanted with *SIAH2* knockout cells were decreased, whereas concurrent deficiency of SIAH2 and DBC1 restored tumor growth to a certain extent (***Figure 5F–H***). Immunohistochemical analysis revealed that the number of Ki67-positive proliferative cells in *SIAH2* knockout-implanted tumors was decreased compared with that in both wildtype and *SIAH2/CCAR2* double-knockout xenografts (***Figure 5I***). Furthermore, the results of EdU assay to detect the proliferation rate of tumor cells of different genotypes were consistent with immunohistochemical analysis. (***Figure 5J and K***). Collectively, these results demonstrate the critical role of the SIAH2-DBC1 axis in promoting tumor progression.

## Correlation of SIAH2 and DBC1 expression with tumor progression in breast cancer

To evaluate whether the mechanism by which the SIAH2-DBC1 axis regulates tumor progression is relevant to human tumorigenesis and tumor progression. We analyzed RNA-sequencing data in TCGA (Cancer Genome Atlas) using TIMER, and the results demonstrated that the expression level of SIAH2 was significantly higher in most tumors than in adjacent normal tissues (***Figure 6A***). In particular, the expression of SIAH2 in breast cancers was higher than that in normal samples (***Figure 6B***). Analyzing the data from TCGA, we found that SIAH2 was positively correlated with tumor stage and number of lymph nodes, indicative of tumor malignancy (***Figure 6C and D***). To further explore the clinical relevance of SIAH2-mediated DBC1 degradation, we analyzed the expression of SIAH2 and DBC1 in breast cancer tissue microarrays of patients. The immunohistochemistry results showed that DBC1

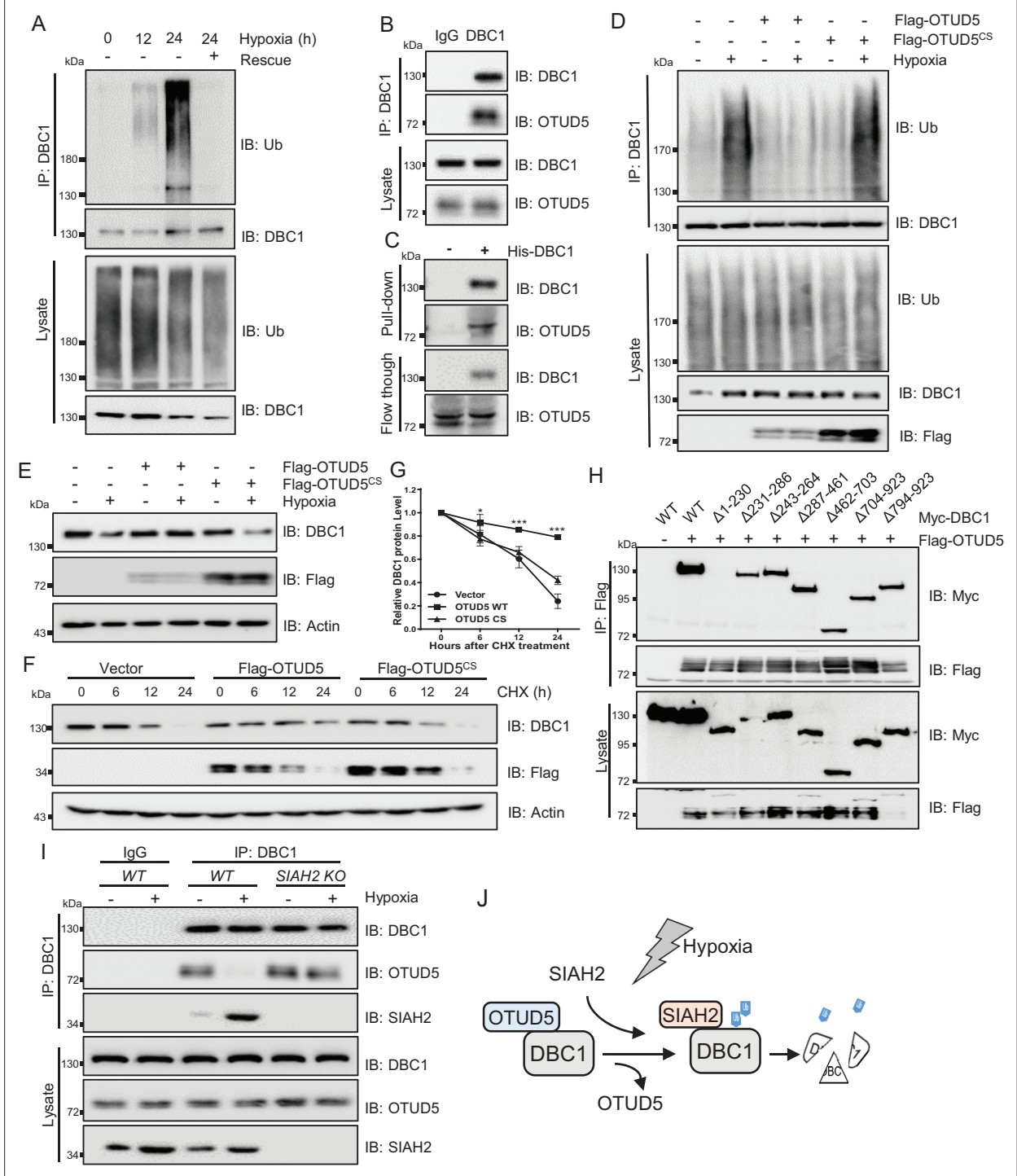

**Figure 4.** OTUD5 interacts with DBC1 and regulates its stability. (**A**) MDA-MB-231 cells were cultured under normoxia or hypoxia for indicated time, then one group was recovered to normoxia for 6 hr, and then treated with MG132 (10 μM) for 6 hr. Cells were harvested, denatured, and lysed for immunoprecipitation with anti-DBC1 antibody. The ubiquitination level of DBC1 was assessed by immunoblotting with anti-Ub antibody. (**B**) MDA-MB-231 cells were collected for immunoprecipitation with anti-DBC1 antibody, and co-immunoprecipitated endogenous OTUD5 was detected by Western blotting with an anti-OTUD5 antibody. (**C**) Purified His-tagged DBC1 proteins were used for His affinity isolation of endogenous OTUD5 of MCF7 cells and blotted with an anti-OTUD5 antibody. (**D**) HeLa cells were transfected with Flag-OTUD5 or the Flag-OTUD5 (C224S) mutant and cultured under normoxia or hypoxia for 24 hr and then treated with MG132 (10 μM) for 6 hr. Cells were harvested, denatured, and lysed for immunoprecipitation with anti-DBC1 antibody. The ubiquitination level of DBC1 was assessed by immunoblotting with anti-Ub antibody. (**E**) HeLa cells were transfected with Flag-OTUD5 or the Flag-OTUD5 (C224S) mutant and cultured under normoxia or hypoxia for 24 hr and then protein level of DBC1 was detected by

*Figure 4 continued on next page*

*Figure 4 continued*

Western blotting. (**F**) HeLa cells were transfected with Flag-OTUD5 or the Flag-OTUD5 (C224S) mutant and exposed to hypoxia for 24 hr, CHX (10 µM) was added for the indicated time, and the cell lysates were subjected to Western blotting analysis of DBC1. (**G**) Quantification of DBC1 protein levels in (**F**) (mean ± SEM from three independent experiments, two-way ANOVA, *p<0.05, ***p<0.001). (**H**) HEK293T cells were co-transfected with full-length or truncated forms of Myc-DBC1 and Flag-OTUD5, and immunoprecipitation was performed with an anti-Flag antibody. Co-immunoprecipitated DBC1 and OTUD5 were detected by Western blotting with anti-Myc and anti-Flag antibodies, respectively. (**I**) The WT and *SIAH2* knockout MDA-MB-231 cells were exposed to normoxia or hypoxia for 24 hr and then treated with MG132 (10 µM) for 6 hr. Immunoprecipitation was performed with an anti-DBC1 antibody. Co-immunoprecipitated endogenous SIAH2 and OTUD5 were detected by Western blotting with anti-SIAH2 and anti-OTUD5 antibodies, respectively. (**J**) Schematic model presenting the substitution interaction of SIAH2 and OTUD5 with DBC1 under hypoxia.

The online version of this article includes the following source data and figure supplement(s) for figure 4:

**Source data 1.** OTUD5 interacts with DBC1.

**Figure supplement 1.** The stability of DBC1 was regulated by OTUD5.

**Figure supplement 1—source data 1.** OTUD5 regualte the deubiquitination of DBC1.

and SIAH2 were negatively correlated in this cohort (*Figure 6E and F*, *Figure 6—figure supplement 1A*), coincident with our finding that DBC1 was the substrate of SIAH2. We also found that in human breast cancer tissue DBC1 expression was reduced in hypoxic regions (*Figure 6G*), and the down-regulation of DBC1 was found to be negatively correlated with clinical stage and the percentage of the Ki67-positive cell population (*Figure 6H and I* and *Supplementary file 2*), further indicating that under pathological conditions SIAH2-mediated DBC1 ubiquitination and degradation are beneficial for tumor progression.

## Discussion

It has been documented that DBC1, a specific nuclear protein containing multifunctional domains, participates in the positive and negative regulation of multiple signaling pathways. Several findings have suggested that DBC1 acts as a natural and endogenous inhibitor of SIRT1, and DBC1 deletion increases the SIRT1-p53 interaction and represses p53 transcriptional activity to inhibit apoptosis and promote tumorigenesis (*Kim et al., 2008*; *Zhao et al., 2008*; *Noh et al., 2013*; *Qin et al., 2015*; *Akande et al., 2019*). Given the importance of DBC1 function, the regulatory mechanisms by which DBC1 protein stability is regulated remain unclear. Here, we showed that the protein level of DBC1 was degraded under hypoxic stress, which was regulated by the E3 ubiquitin ligase SIAH2 and deubiquitinase OTUD5. Additionally, knockout of *SIAH2* promoted apoptosis and reduced cell migration and tumor proliferation, whereas double knockout of *CCAR2* and *SIAH2* partially rescued cell proliferation and tumorigenesis. Importantly, DBC1 is negatively correlated with SIAH2 expression levels in human breast tumors, suggesting that the SIAH2–DBC1 axis pathway may play a key role in human breast cancer. These results provide novel insights into the metastatic mechanisms of human breast cancer.

An abundance of evidence has shown that SIAH2 mediates the ubiquitination and degradation of substrates, including PI3K, LATS2, Spry2, ACK1, and TYK2 (*Chan et al., 2017*; *Ma et al., 2016*; *Buchwald et al., 2013*; *Nadeau et al., 2007*; *Müller et al., 2014*), in response to hypoxic stress to modulate multiple signaling pathways, such as the Hippo pathway, Ras signaling pathway, and STAT3 pathway. In general, DBC1 activity is regulated by multiple post-translational modifications, including phosphorylation (*Magni et al., 2015*; *Zannini et al., 2012*), acetylation (*Zheng et al., 2013*; *Rajendran et al., 2019*) and sumoylation (*Park et al., 2014*). However, it remains unclear whether DBC1 function can be regulated by ubiquitination modification. Our group demonstrated for the first time that SIAH2 can also ubiquitinate DBC1 at Lys287 by binding to the N-terminus of DBC1, resulting in DBC1 degradation by the ubiquitin–proteasome pathway, and that deletion of *SIAH2* will block DBC1 degradation in response to hypoxic stress. Collectively, we identify a novel mechanism by which SIAH2 regulates DBC1 protein stability in response to hypoxic stress, contributing to tumorigenesis and tumor progression.

It has been reported that DBC1 cooperates with SIRT1 to mediate different physiological functions within mammalian cells. For example, stimulation of DBC1 transcription inhibits SIRT1 activity, contributing to TGF-β-induced epithelial–mesenchymal transition (*Chen et al., 2021*). Additionally, SIRT7 represses DBC1 transcription to promote thyroid tumorigenesis by binding to the promoter of DBC1 (*Li et al., 2019*). Recently, DBC1 was found to play a role in upregulating glucose homeostasis-related

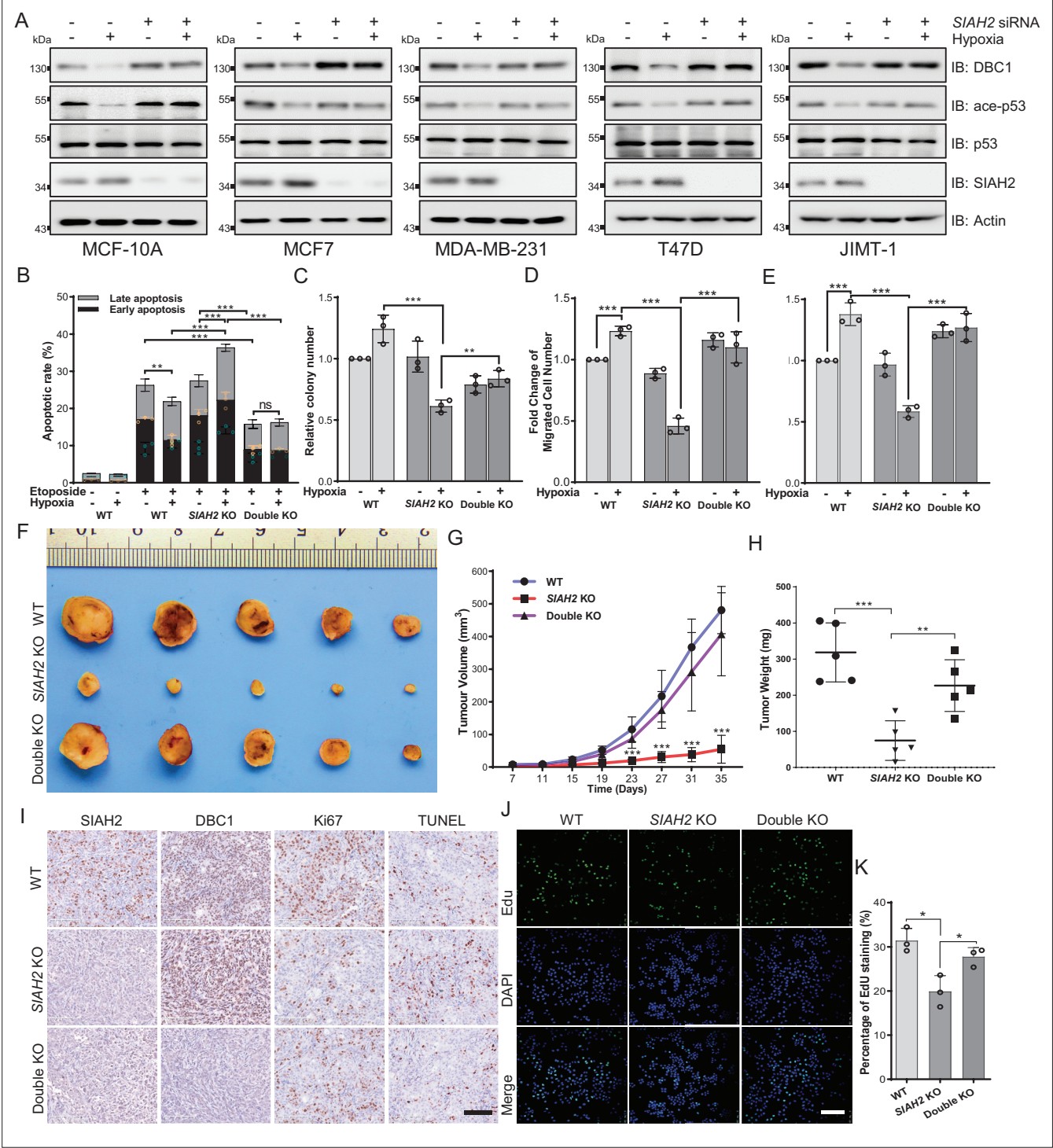

**Figure 5.** Hypoxia regulates tumor progression via the SIAH2-DBC1 axis. (**A**) SIAH2 siRNA was transfected into indicated cell lines. The cells were cultured under normoxia or hypoxia for 24 hr and then cell lysates were analyzed by immunoblotting using the indicated antibodies. (**B**) Apoptosis assay analysis of wildtype, *SIAH2* knockout, and *SIAH2/CCAR2* double-knockout MDA-MB-231 cells under normoxia or hypoxia. (**C**) Colony formation analysis of wildtype, *SIAH2* knockout, and *SIAH2/CCAR2* double-knockout MDA-MB-231 cells under normoxia or hypoxia. Colony numbers were quantified. (**D**) Transwell analysis of wildtype, *SIAH2* knockout, *and SIAH2/CCAR2* double-knockout MDA-MB-231 cells under normoxia or hypoxia. (**E**) Scratching analysis of wildtype, *SIAH2* knockout, and *SIAH2/CCAR2* double-knockout MDA-MB-231 cells under normoxia or hypoxia. (**F–H**) Tumor images (**F**), tumor growth curves (**G**), and tumor weight (**H**) from mice subcutaneously injected with *SIAH2* knockout or *SIAH2/CCAR2* double-knockout MDA-MB-231 cells. (**I**) Immunohistochemical analysis of *SIAH2* knockout and *SIAH2/CCAR2* double-knockout xenograft tumor tissues with the indicated antibodies. Scale bars, 100 μm. (**J**) EdU proliferation analysis of different tumor cells isolated from *SIAH2* knockout or *SIAH2/CCAR2* double-

*Figure 5 continued on next page*

*Figure 5 continued*

knockout xenografts. Scale bars, 200 µm. (**K**) The percentage of EdU-positive cells from (**J**) was quantified. Data shown are representative of at least three independent experiments. Similar results were found in each experiment. All data are mean ± SEM; *p<0.05, **p<0.01, ***p<0.001. Statistical significance was analyzed by using the two-tailed unpaired Student's *t*-test.

The online version of this article includes the following source data and figure supplement(s) for figure 5:

**Source data 1.** Source data of Hypoxia regulates tumor progression via the SIAH2-DBC1 axis.

**Figure supplement 1.** Hypoxia regulates DBC1 stability to affect tumor progression.

genes, which are implicated in type 2 diabetes pathogenesis (*Basu et al., 2020*). Strikingly, our findings reveal that SIAH2-mediated DBC1 ubiquitination under hypoxia regulates cell proliferation and tumorigenesis. We also found that the decrease in p53 acetylation was accompanied by DBC1 degradation in hypoxia zone of breast tumor.

In general, ubiquitination is a dynamic and reversible process cooperatively regulated by E3 ligases and DUBs (*Li and Reverter, 2021*). In this study, we identified that the deubiquitinase OTUD5 could specifically cleave the polyubiquitin chains of DBC1 once hypoxic stress was removed. If OTUD5 was overexpressed under hypoxia, the ubiquitination and degradation of DBC1 would be inhibited. Interestingly, we further confirmed that SIAH2 and OTUD5 competitively bind to DBC1 at the same N-terminal region, and hypoxia promotes the interaction of DBC1 with SIAH2 rather than OTUD5, resulting in ubiquitination and degradation of DBC1 to promote tumor progression. It is well known that OTUD5 controls cell survival and cell proliferation by deubiquitinating substrates (*Zhang et al., 2021*; *Guo et al., 2021*; *Cho et al., 2021*; *Li et al., 2020*; *de Vivo et al., 2019*; *Park et al., 2015*; *Luo et al., 2013*; *Kayagaki et al., 2007*). Based on our findings and others', we speculate that the activation of OTUD5-mediated DBC1 deubiquitination may suppress tumor growth, which may provide a novel target for clinical tumor therapy. Moreover, we still need to further investigate the details of how the OTUD5-DBC1 complex dissociates under hypoxia to modulate DBC1 stability. There are also some limitations in the study; as we have shown that DBC1 reduction only takes place in the hypoxic regions in a tumor, the tumor microenvironment could have more factors besides hypoxia stress. However, the tumor microenvironment could have more factors besides hypoxia stress. In addition, the expression changes of SIAH2 and OTUD5 under hypoxia require more studies to explain.

Taken together, our study revealed that hypoxia stimulates SIAH2 to ubiquitinate DBC1 and inhibit OTUD5-mediated DBC1 deubiquitination, resulting in DBC1 degradation through the ubiquitin–proteasome pathway. Our results further address the importance of the SIAH2-DBC1 axis in promoting tumor cell survival and migration. In conclusion, we uncovered a complete and detailed dynamic regulatory mechanism by which DBC1 protein ubiquitination and stability are regulated. It is of great significance to deeply understand the function of SIAH2 and DBC1 and the role of hypoxia in regulating tumorigenesis and tumor progression, which provides a solid theoretical basis for cancer treatment.

## Materials and methods
### Construction of plasmids
The expression plasmids for human SIAH2, DBC1, OTUD5, and ubiquitin were generated by amplifying the corresponding cDNA by PCR and cloning it into pcDNA4-TO-Myc-His-B, pCMV-3xFLAG, pEGFP-C1, pRK5-HA, pGEX4T1, or PET28a expression vectors. Site-specific mutants and special domain deletion mutants were generated using TransStart FastPfu DNA Polymerase (Transgene, Cat# AP221-11) according to the manufacturer's protocols.

### Cell culture and transfection
HEK293T cells, HeLa cells, MDA-MB-231 cells, MCF7 cells, SH-SY5Y cells, U2OS cells, SW480 cells, HCT116 cells, PC3 cells, and HepG2 cells were sourced from ATCC, maintained Mycoplasma-negative culture, and cells were regularly tested for Mycoplasma contamination. Cells were cultured in GIBCO Dulbecco's modified Eagle's medium (DMEM) supplemented with 10% fetal bovine serum (FBS, Lanzhou Bailing) and 100 units/mL penicillin and 100 mg/mL streptomycin in a 5% $CO_2$ incubator at 37°C. All cells were cultured in an atmosphere of 5% $CO_2$ at 37°C, and hypoxia was induced by

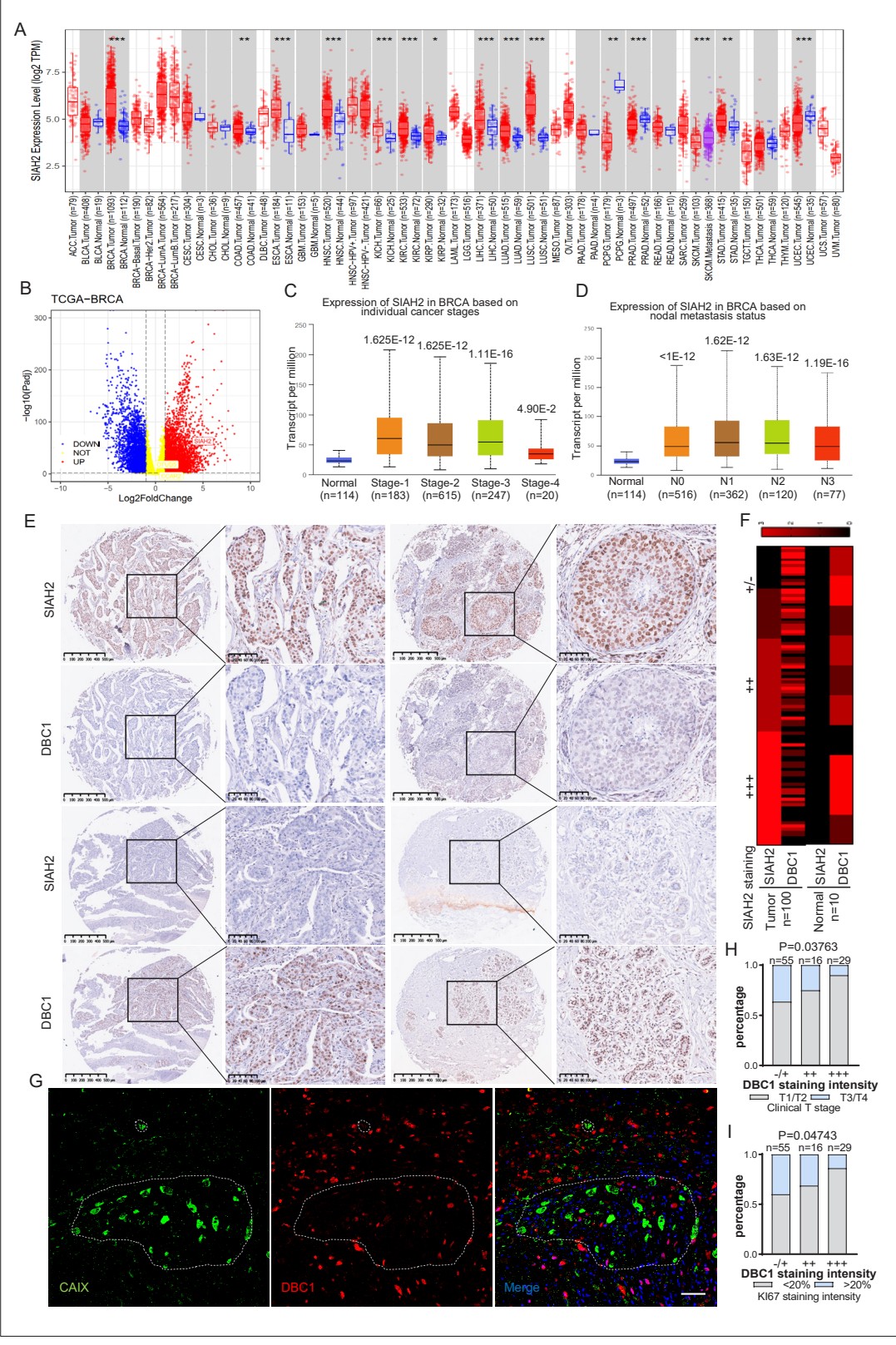

**Figure 6.** Correlation of SIAH2 and DBC1 expression with tumor progression in BRCA. (**A**) SIAH2 expression level in different human cancers from TCGA data in TIMER. *p<0.05, **p<0.01, ***p<0.001. (**B**) Volcano plot analysis of SIAH2 DBC1 and OTUD5 expression levels in breast cancer. (**C**) Relative expression level of SIAH2 in tumor stage (stages 1, 2, 3, 4, or 5) from BRCA patients. (**D**) Relative expression level of SIAH2 in nodal metastasis status

*Figure 6 continued on next page*

*Figure 6 continued*

(normal, N0, or N1) from BRCA patients. (**E**) Immunohistochemistry staining analysis of the expression levels of SIAH2 and DBC1 in a series of breast cancer patient tissue microarrays. Scale bars, (left) 200 μm; (right) 100 μm. (**F**) Heatmap of the expression levels of SIAH2 and DBC1 in human normal breast tissues and breast tumor tissues. (**G**) Human breast cancer tissues were stained with DAPI (blue) together with anti-CAIX (green) and anti-DBC1 (red) antibodies. Scale bars, 50 μm. (**H**) Statistical analysis of correlations between DBC1 expression level and clinical T stage. (**I**) Statistical analysis of correlations between DBC1 expression level and Ki67-positive stage.

The online version of this article includes the following figure supplement(s) for figure 6:

**Figure supplement 1.** The expression levels of SIAH2 and DBC1 were negative correlation in BRCA.

culturing cells in a hypoxia chamber (Billups-Rothenberg) flushed with a mixed gas of 1% $O_2$, 5% $CO_2$, and 94% $N_2$. For HeLa and HEK293T cells, plasmids were transfected with polyethyleneimine (PEI; Polysciences, Cat# 23966) according to the manufacturer's protocols. For MDA-MB-231 and MCF7 cells, plasmids were transfected with Lipofectamine 3000 (Invitrogen, Cat# L3000015) according to the manufacturer's instructions.

## CRISPR/Cas9-mediated gene knockout

To generate SIAH2- and DBC1- knockout MDA-MB-231 and MCF7 cells, oligonucleotides were cloned into LentiCRISPR. The backbone plasmid was co-transfected with lentiviral packaging plasmids psPAX2 and pMD2.G for lentivirus production. The oligonucleotide pair used was as follows: *SIAH2* (F: 5'-CACCGGCTGCAGGGTTTATTAGCGC-3' and R: 5'-AAAC GCGCTAATAAACCCTG CAGC C-3'), *CCAR2* (F: 5'-CACCG TGGTTTGCTCACTCCTCCTG-3' and R: 5'-AAAC CAGGAGGA GTGAGCAAACCA C-3'). Knockout of the respective genes was further confirmed by sequencing the edited genomic regions after PCR. The loss of the protein was verified by Western blotting.

## Co-immunoprecipitation and pull-down assay

After the described treatment, cells were collected and lysed in 0.8 mL IP lysis buffer (150 mM NaCl, 20 mM Tris, pH 7.4, 1 mM EGTA, 1 mM EDTA, 1% NP-40, 10% glycerol) containing protease inhibitors (1:100, Roche) for 45 min on a rotor at 4°C. After centrifugation at 12,000 × $g$ for 10 min, the supernatant was immunoprecipitated with 1.5 μg of specific antibodies overnight at 4°C. Protein A/G agarose beads (15–30 μL, Santa Cruz) were washed and then added for another 2 hr. The precipitants were washed seven times with wash buffer, and the immune complexes were boiled with loading buffer and analyzed. GST-SIAH2 and His-DBC1 proteins were generated in *Escherichia coli* and incubated in vitro overnight. Then, 25 μL GST beads (GE Healthcare) were added to the system for 2 hr on a rotor at 4°C.

## Immunoprecipitation and mass spectrometric analysis of SIAH2-associated proteins

Twelve 10 cm dishes of HEK293T cells were transiently transfected with Flag–SIAH2$^{RM}$. Then, 24 hr later, cells were collected and lysed in lysis buffer followed by immunoprecipitation with anti-Flag antibody overnight at 4°C. The precipitants were extensively washed with lysis buffer and then boiled with SDS loading buffer and subjected to SDS–PAGE. Gels with total proteins were excised, followed by in-gel digestion and analysis by LC-MS/MS.

## Ubiquitination assay

### In vivo ubiquitination assays

Cells were transiently transfected with plasmids expressing HA-ubiquitin or special site mutations and Myc-DBC1 together with Flag-SIAH2 or Flag-SIAH2$^{RM}$ 24 hr after transfection. Cells were treated with 10 μM MG132 (Selleck, S2619) for 6 hr before collection. Cells were washed with cold PBS and then lysed in 200 μL of denaturing buffer (150 mM Tris-HCl, pH 7.4, 1% SDS) by sonication and boiling for 15 min. Lysates were made up to 1 mL with regular lysis buffer and immunoprecipitated with 2 μg anti-c-Myc antibody at 4°C overnight, washed three times with cold lysis buffer, and then analyzed by SDS–PAGE. For the DBC1 endogenous ubiquitination assay, cells were treated with 10 μM MG132 for 6 hr before collection. Lysates were immunoprecipitated using 2 μg anti-Ub antibody or anti-DBC1 antibody and subjected to ubiquitination analysis by Western blotting with anti-DBC1 or anti-ubiquitin

antibody. In vitro ubiquitination assays were carried out in ubiquitination buffer (50 mM Tris, pH 7.4, 5 mM MgCl$_2$, 2 mM dithiothreitol) with human recombinant E1 (100 ng, Upstate), human recombinant E2 UbcH5c (200 ng, Upstate), and His-tagged ubiquitin (10 μg, Upstate). GST-SIAH2, GST-SIAH2$^{RM}$, and His-DBC1 were expressed and purified from *E. coli* BL21 cells. Also, 2 μg of GST, GST-SIAH2, or GST-SIAH2$^{RM}$ protein was used in the corresponding ubiquitination reactions. Reactions (total volume 30 μL) were incubated at 37°C for 2 hr and subjected to ubiquitination analysis by Western blotting using anti-DBC1 antibody.

## Western blotting

Cells were collected and washed with PBS and then lysed in 1% SDS lysis buffer or NP-40 lysis buffer (150 mM NaCl, 20 mM Tris, pH 7.4, 1 mM EGTA, 1 mM EDTA, 1% NP-40, 10% glycerol) containing protease inhibitors (1:100, Roche) for 45 min on a rotor at 4°C. After centrifugation at 12,000 × *g* for 10 min, the supernatant was boiled with loading buffer. Cell lysates containing equivalent protein quantities were subjected to 6 or 10% SDS–PAGE, transferred to nitrocellulose membranes, and then incubated with 5% milk for 2 hr at room temperature. Then, the membranes were probed with related primary antibodies at 4°C, followed by the appropriate HRP-conjugated secondary antibodies (KPL). Immunoreactive bands were checked with a chemiluminescence kit (En-green Biosystem) and visualized with a chemiluminescence imager (JUNYI). The following antibodies were used: Flag-M2 (1:2000, Sigma), Myc (1:1000, Santa Cruz Biotechnology), HA (1:1000, Santa Cruz Biotechnology), GFP (1:1000, Santa Cruz Biotechnology), His (1:1000, Santa Cruz Biotechnology), GST (1:1000, Santa Cruz Biotechnology), anti-ACTIN (1:10,000, Sigma), anti-mono- and polyubiquitinated conjugate monoclonal antibodies (FK2) (1:1000, Enzo Life Sciences), SIAH2 (1:500 Proteintech), OTUD5 (1:1000 Proteintech and CST), DBC1 (1:1000 Proteintech, CST, and Abcam), SIRT1 (1:1000 Abcam), p53 (1:1000 Abcam), acetylated-K382-p53 (1:1000 Abcam), HIF1α (1:1000 Proteintech), and GSK3β (1:1000 Proteintech).

## Immunofluorescence microscopy

Cells were fixed with 4% paraformaldehyde (Dingguo Changsheng Biotechnology) at 37°C for 30 min, then permeabilized with 0.2% Triton X-100 (Shanghai Sangon Biotech, TB0198) at 4°C for 10 min. After blocking in goat serum for 2 hr, cells were incubated with the indicated antibodies for 2 hr at room temperature or overnight at 4°C, washed with 0.05% Triton X-100, and stained with FITC- or CY5-conjugated secondary antibodies (Invitrogen) for 1 hr at room temperature. Cell images were captured with TCS SP5 Leica confocal microscope. The following antibodies were used: CAIX (1:100, Proteintech, 66243-1-Ig); DBC1 (1:1000, CST, 5693); and DAPI (300 nM, Invitrogen, D21490).

## Flow cytometry

After the described treatment, cells were collected and washed twice with prewarmed FBS-free DMEM and then stained with PI and Annexin V (Thermo Fisher) for 20 min at 37°C. After staining, cells were washed twice with prewarmed PBS for analysis with a flow cytometer (BD Calibur).

## RNA extraction and quantitative real-time PCR assay

RNA samples were extracted with TRIzol reagent (Sigma T9424), reverse transcription PCR was performed with a Reverse Transcriptase kit (Promega A3803), and real-time PCR was performed using Powerup SYBR Green PCR master mix (A25743) and a Step-One Plus real-time PCR machine (Applied Biosystems). Human actin expression was used for normalization. Quantitative real-time PCR assay (data are the mean ± SEM of three experiments, ns means not significant, *p<0.05, **p<0.01, ***p<0.001).

## RNA-sequencing analysis

RNA was extracted in biological triplicates using the miRNeasy Mini kit (QIAGEN) according to the manufacturer's instructions. RNA degradation and contamination were monitored on 1% agarose gels. RNA purity was checked using a NanoPhotometer spectrophotometer (Implen, CA). RNA integrity was assessed using the RNA Nano 6000 Assay Kit of the Bioanalyzer 2100 system (Agilent Technologies, CA). RNA quality control was performed using a fragment analyzer and standard or high-sensitivity RNA analysis kits (Labgene; DNF-471-0500 or DNF-472-0500). RNA concentrations were measured using the Quanti-iTTM RiboGreen RNA assay Kit (Life Technologies/Thermo Fisher Scientific). A total

of 1000 ng of RNA was utilized for library preparation with the NEBNext Ultra RNA Library Prep Kit for Illumina (NEB, USA) following the manufacturer's recommendations, and index codes were added to attribute sequences to each sample. Poly-A + RNA was sequenced with a HiSeq SBS Kit v4 (Illumina) on an Illumina HiSeq 2500 using protocols defined by the manufacturer.

Raw data (raw reads) in fastq format were first processed through in-house Perl scripts. In this step, clean data (clean reads) were obtained by removing reads containing adapters, reads containing poly-N and low-quality reads from raw data. At the same time, the Q20, Q30, and GC contents of the clean data were calculated. All downstream analyses were based on clean data with high quality. Reference genome and gene model annotation files were downloaded from the genome website directly. The index of the reference genome was built using Hisat2 v2.2.1, and paired-end clean reads were aligned to the reference genome using Hisat2 v2.2.1. We selected Hisat2 as the mapping tool because Hisat2 can generate a database of splice junctions based on the gene model annotation file and thus a better mapping result than other nonsplice mapping tools. The mapped reads of each sample were assembled by StringTie (v2.1.4) (Mihaela Pertea et al. 2015) in a reference-based approach. StringTie uses a novel network flow algorithm as well as an optional de novo assembly step to assemble and quantitate full-length transcripts representing multiple splice variants for each gene locus. Differential expression analysis of two conditions/groups (two biological replicates per condition) was performed using the DESeq2 R package (1.30.1). DESeq2 provides statistical routines for determining differential expression in digital gene expression data using a model based on the negative binomial distribution. The resulting p-values were adjusted using Benjamini–Hochberg's approach for controlling the false discovery rate. Genes with an adjusted p-value of 0.05 and absolute log2Fold-Change of 0.5 were set as the threshold for significantly differential expression. Sequencing data have been deposited in GEO under accession codes GSE193133.

## Colony formation assay

MDA-MB-231 and MCF7 cells were serum-starved for 24 hr, and 500 cells/well were seeded on 6-well plate in triplicate. Cells were cultured under normoxia or hypoxia and incubated until formatted 50 cells by colony (approximately 10–15 days). Then, colonies were also dyed using crystal violet and the number of colonies was counted by using the software ImageJ.

## Transwell assay

MDA-MB-231 and MCF7 cells were serum-starved for 24 hr, and $5 \times 10^4$ cells were seeded on 24-well plate of 8.0 µm pore-size diameter polycarbonate (PC) membrane transwell inserts (Corning, Cat#3422, USA). After 24 hr, the non-migrated cells were removed from the insert with a cotton swab. The migrated cells were fixed for 10 min in 3.7% (v/v) formaldehyde in PBS before staining with 0.1% crystal violet for 15 min, followed by washing with PBS. Images were taken and the crystal violet-stained cells were counted.

## Scratch/wound healing assay

MDA-MB-231 and MCF7 cells were plated to confluence in 6-well culture plates containing DMEM medium (with 1% FBS and 1% penicillin/streptomycin). Using a 10 µL pipette tip, three horizontal scratches were made across the diameter of the well. In addition, on the bottom/back of each well, a line perpendicular to the scratches was drawn with a permanent marker. The monolayer was washed one time with PBS to remove any floating cells generated by the scratching process and fresh media applied. Using an inverted microscope, photo at the intersection of each cell scratch and the line on the bottom of the plate were captured immediately (six images per well) and used as a reference point for the 24 hr time point to determine percentage scratch coverage.

## Xenograft tumorigenesis study

All mouse experiments were approved by the Institutional Animal Care and Use Committee (2022-SYDWLL-000353) at the College of Life Sciences at Nankai University. MDA-MB-231 breast cancer cells ($1 \times 10^6$ in 200 µL PBS) were injected subcutaneously into the armpit of 6- to 8-week-old female BALB/c nude mice. Tumor size was measured every 3–5 days 1 week after the implantation, and tumor volume was also analyzed by using the formula $V = 0.5 \times L \times W^2$ (V, volume; L, length; W, width). The mice were then sacrificed, and the subcutaneous tumors were surgically removed, weighed, and

photographed. No statistical method was used to predetermine the sample size for each group. The experiments were not randomized.

## Tissue microarrays and immunohistochemistry

The breast cancer tissue microarrays were purchased from US Biomax. These tissue microarrays consisted of 100 analyzable cases of invasive breast carcinoma and 10 analyzable cases of normal breast tissue. For antigen retrieval, the slides were rehydrated and then treated with 10 mM sodium citrate buffer (pH 6.0) heated for 3 min under pressure. The samples were treated with 3% $H_2O_2$ for 15 min to block endogenous peroxidase activity and then blocked with 5% goat serum for 1 hr at room temperature. Then, the tissues were incubated with the indicated antibodies at 4°C overnight, followed by incubation with HRP-conjugated secondary antibody for 1 hr at room temperature. Immunoreactive signals were visualized with a DAB Substrate Kit (MaiXin Bio). Protein expression levels in all the samples were scored on a scale of four grades (negative, +, ++, +++) according to the percentage of immunopositive cells and immunostaining intensity. The $\chi^2$ test was used for analysis of statistical significance. The following antibodies were used for immunohistochemistry: Ki67 (1:200, Abcam, Cat# ab16667, RRID:AB_302459), SIAH2 (1:40, Novus Biologicals, NB110-88113), and DBC1 (1:100, Proteintech, 22638-1-AP).

## Quantification and statistical analysis

For quantitative analyses of Western blots, real-time PCR results, flow cytometry data, or cell numbers, values were obtained from three independent experiments. The quantitative data are presented as the means ± SEM. Student's *t*-test was performed to assess whether significant differences existed between groups. Multiple comparisons were performed with one-way ANOVA and Tukey's post-hoc test. For correlations between DBC1 protein level and clinical T stage and Ki67 staining intensity in clinical samples, statistical significance was determined using the $\chi^2$ test. p-Values <0.05 were considered statistically significant. The significance level is presented as *p<0.05, **p<0.01, and ***p<0.001. 'ns' indicates that no significant difference was found. All analyses were performed using Prism 8.0 (GraphPad Software, Inc, La Jolla, CA).

## Acknowledgements

We are grateful to Dr. Wenhui Zhao for providing us the OTUD5 plasmids at the Center for Peking University Health Science Center. We are also grateful to Dr. Changhai Tian from the University of Kentucky College of Medicine for critical reading of the manuscript.

## Additional information

### Funding

| Funder | Grant reference number | Author |
|---|---|---|
| National Key Research and Development Program of China | 2019YFA0508603 | Yushan Zhu |
| National Natural Science Foundation of China | 32030026 | Yushan Zhu |
| National Natural Science Foundation of China | 91849201 | Quan Chen |
| National Natural Science Foundation of China | 91754114 | Yushan Zhu |
| National Natural Science Foundation of China | 31790404 | Quan Chen |

The funders had no role in study design, data collection and interpretation, or the decision to submit the work for publication.

## Author contributions
Qiangqiang Liu, Conceptualization, Resources, Data curation, Investigation, Writing – original draft; Qian Luo, Investigation, Methodology, Writing – original draft; Jianyu Feng, Biao Ma, Hongcheng Cheng, Tian Zhao, Hong Lei, Chenglong Mu, Linbo Chen, Yuanyuan Meng, Jiaojiao Zhang, Yijia Long, Jingyi Su, Guo Chen, Yanjun Li, Gang Hu, Xudong Liao, Methodology; Yanping Zhao, Software, Methodology; Quan Chen, Funding acquisition, Investigation, Methodology; Yushan Zhu, Conceptualization, Supervision, Funding acquisition, Writing – original draft, Project administration, Writing – review and editing, Validation, Visualization

## Author ORCIDs
Yushan Zhu http://orcid.org/0000-0002-5648-0416

## Ethics
Mice were maintained in the animal core facility of College of Life Sciences, Nankai University, Tianjin, China. All experiments involving animals were reviewed and approved by the Animal Care and Use Committee of Nankai University and were performed in accordance with the university guidelines (no. 2022-SYDWLL-000353).

## Decision letter and Author response
Decision letter https://doi.org/10.7554/eLife.81247.sa1
Author response https://doi.org/10.7554/eLife.81247.sa2

# Additional files

## Supplementary files
• MDAR checklist

• Supplementary file 1. Identification of DBC1 as a SIAH2-associated protein. List of representative proteins identified by SIAH2 Co-IP/MS and the number of peptides for each protein-identified peptide are indicated.

• Supplementary file 2. Patient characteristics based on DBC1 expression. The relationship between the clinicopathological characteristics of breast cancer patients and the expression level of DBC1.

## Data availability
Sequencing data have been deposited in GEO under accession codes GSE193133. All data generated or analysed during this study are included in the manuscript and supporting file; Source Data files have been provided for Figures 1-6 and Figure S1-5.

The following dataset was generated:

| Author(s) | Year | Dataset title | Dataset URL | Database and Identifier |
|---|---|---|---|---|
| Qiangqiang Liu, Yanping Zhao | 2022 | Next Generation Sequencing Facilitates Quantitative Analysis of Wild Type and CCAR2-/- MDA-MB-231 cells Transcriptomes: Liu Q, 2021 | http://www.ncbi.nlm.nih.gov/geo/query/acc.cgi?acc=GSE193133 | NCBI Gene Expression Omnibus, GSE193133 |

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
