## [Editor Report]

This is a study with relatively convincing data on examining the dynamic regulation of hypoxia regulating SIAH2/OTUD5 interaction with DBC1 by regulating DBC1 stability. Therefore, DBC1, by regulating p53 signaling, would affect breast cancer phenotype in vivo.

---

## [Decision Letter]

**Decision letter after peer review:**

[Editors’ note: the authors submitted for reconsideration following the decision after peer review. What follows is the decision letter after the first round of review.]

Thank you for submitting the paper "Hypoxia Dynamically Regulates DBC1 Ubiquitination and Stability by SIAH2 and OTUD5 in Breast Cancer Progression" for consideration by *eLife*. Your article has been reviewed by 3 peer reviewers, one of whom is a member of our Board of Reviewing Editors, and the evaluation has been overseen by a Senior Editor.

Unfortunately, numerous significant concerns were raised by each of the 3 reviewers. In considering the sum total of their comments, we are sorry to say that, after consultation with the reviewers, we have decided that this work will not be considered further for publication by *eLife*. The full details are provided below, but the reviewers raised significant concerns about the exact characterization of DBC1 signaling in breast cancer, the underlying mechanism by which hypoxia regulates this signaling axis and the physiological relevance of findings in breast cancer patients. Without a higher level of enthusiasm, we do not feel this paper would fare well during additional rounds of review at *eLife*. We hope these comments are useful to you as you consider your next steps for this manuscript.

*Reviewer #1 (Recommendations for the authors):*

This study, in general, provides the compelling biochemistry and functional data on examining how SIAH2/OTUD5 may regulate DBC1 protein stability in breast cancer under hypoxia. As a result of DBC1 downregulation of hypoxia, breast cancer may display more malignant phenotype. The study is well designed. However, there are a few things that need to be addressed as suggested below:

1. The functional role of DBC1 in breast cancer needs to be characterized in details. For example, the expression of DBC1 in different subtypes of breast cancer. The phenotype of DBC1 overexpression and KO in breast cancer needs to be characterized in details.

2. Throughout the study, two main cell lines are used Hela and MDA-MB-231. Hela cells may display diminished p53 protein due to HPV expression while MDA-MB-231 cells contain mutant p53. It is unclear whether Hypoxia, by downregulating DBC1 level, would lead to further decrease of p53 activity in these cells. This would need to be clarified and reconciled. In addition, some other breast cancer cell lines with WT p53 (at least from in vitro) would need to be added to further strengthen the conclusion.

3. The molecular mechanism by which hypoxia regulates SIAH2/OTUD5 interaction with DBC1 needs more detailed analysis and discussion. Is HIF involved in this process?

*Reviewer #2 (Recommendations for the authors):*

This study reports on the control of DCB1 by the ubiquitin ligase Siah2 under hypoxia growth conditions in breast cancer, while the deubiquitinating enzyme OTUD5 limits DCB1 ubiquitination and thereby prolongs its stability under normoxia.

The following are some of the key points raised for authors considerations:

It is unclear what underlies the control of DCB1 stability under hypoxia (and not under other conditions).

It would be important to determine what is the physiological significance of DCB1 loss under hypoxia for breast cancer cells.

This work focuses on breast cancer, and it is necessary to better establish relevance in numerous breast cancer (all types?) as opposed to other tumor types.

The authors claim for dynamic regulation of DCB1 stability, but it is unclear what determines the dynamics.

Experiments are performed largely under one condition and not another; complementing the assessment for hypoxia with normoxia conditions and vice versa is needed for number of studies shown.

In this study, Liu et al. report on post-translational regulation of DBC1 by the ubiquitin ligase Siah2 and the DUB OTUD5, which occurs during hypoxia. The separate roles of Siah2 and DBC1 in tumor progression have been shown in previous publications. Here the authors define the role of Siah2 in regulating DBC1 stability in breast cancer cells during hypoxia. While interesting, several points need to be experimentally addressed before further evaluation of this study as a candidate for publication in *eLife* should take place.

– Why is Siah2 more effective in degrading DBC1 under hypoxia? The title and elsewhere in this manuscript authors claims for dynamic regulation of DBC1 stability. What underlies this dynamic regulation?

– The work suggests that the Siah2 effect on DBC1 occurs in breast cancer cells, but most of the data is based on studies performed in either 293 or HeLa cells. Work on breast cancer lines is primarily limited to a single cell line.

– Are the effects reported here also seen in breast cancer patients?

– Much of the data relies on overexpression. Authors ought to demonstrate changes in endogenous proteins and their physiological relevance (when does it happen and why).

– OTUD5 is shown to regulate DBC1 under normoxic (and not hypoxic?) conditions. Why?

– All figure legends and Results sections lack details. It is not clear how the experiments were carried out.

Additional comments:

Figure 1: the authors assess the overlap of differentially expressed genes between hypoxia and DBC1 deletion (in normoxia). DEG needs to be compared under the same (hypoxic) conditions. Can overexpression of DBC1 rescue phenotypes be seen under hypoxia?

Figure 2A: the authors overexpress several E3 ubiquitin ligases to determine possible candidates responsible for DBC1 protein degradation. However, the level of expression of each E3 ligase (mRNA level) is missing, making it difficult to compare the efficiency of each E3 ligase in degrading DBC1. HUWE1 also seems to be as efficient as Siah2 in degrading DBC1, but it is not mentioned in the Results section. Why? The experiment is carried out in normoxia rather than hypoxia; why? Does any UBL tested exhibit different efficiency for DBC1 interaction and degradation under hypoxia?

Figure 3: The authors should demonstrate, experimentally, why Siah2 dependent degradation of DBC1 is primarily claimed to be under hypoxia.

Figure 3H: the authors compared DBC1 protein levels in WT and Siah2 KO cells using MDA-MB-231 cells. It would be informative to compare the level of DBC1 between WT and Siah2 KO in the same Western and demonstrate possible changes in the level of DBC1 under normoxia. Additional breast cancer cell lines (and non-transformed breast cultures) should be studied.

Figure 3K: It would be informative to perform CHX chase under normoxia, compared with CHX done under hypoxia. These assessments need to be extended to include additional breast cancer cell lines.

Figure 4: the authors should study OTUD5 binding to DBC1 under hypoxia compared with normoxia to substantiate the claim for preferred interaction under normoxia. Does OTUD5 expression change in breast cancer patients, compared to healthy donors or between cancerous and normal tissues? Does it correlate with patient survival? The screening of DUBS to identify the candidate responsible for deubiquitination of DBC1 is not clear. Sup Figure 4 A shows the level of expression of the different DUBS but does not reveal why the authors chose to focus on OTUD5.

Figure4I: the authors should add Siah2 KO cells to their analysis; it is important to determine whether OTUD5 and Siah2 compete for binding to DBC1.

Figure 5B, D, F: the authors should address whether Siah2 KO can promote apoptosis, colony formation assay, and invasion under normoxia, not only in hypoxia.

Figure 5M: the authors should check for possible changes in apoptosis, in addition to shown data for proliferation.

All figure legends and Results sections lack details. It is not clear how the experiments were carried out. (for example, Figure 1A: how long were the cells kept in hypoxia when RNAseq was performed? The authors should define the conditions used to reach hypoxia in the results and Figure legends. Condition of 1% O2 is briefly noted in the Methods section, but not in results or legends.

*Reviewer #3 (Recommendations for the authors):*

The authors sought to identify how the breast cancer tumour suppressor DBC1 (CCAR2) is post-translationally regulated. This came from transcriptomic analysis of MB231 cells in hypoxia revealed decreased SIRT1 activity, without changes in protein levels. Instead known regulator DBC1 displayed reduced protein expression in hypoxia with no changes in mRNA. Through targeted screen of hypoxia associated E3 ligases and IP-MS of SIAH2, the authors identify interaction between DBC1 and SIAH2. They comprehensively map the interaction between DBC1 and SIAH2 and identify the residue ubiquitinated. Upon 6h of reoxygenation the ubiquitination of DBC1 is decreased. OTUD5 was identified as the DUB that deubiquitinates DBC1. The authors propose that in normoxia, OTUD5 is bound to DBC1 which is displaced by SIAH2 upon activation by hypoxia, however despite the comprehensive study of the interaction between DBC1 this is not fully supported by the data. SIAH2 Knockout reduced colony formation, cell migration and growth in a xenograft tumour model which could all be partially rescued by knockout of DBC1. SIAH2 expression was elevated in breast cancer and correlated with cancer stage. Through staining of a TMA the authors suggest a negative correlation between DBC1 and SIAH2 protein expression however this is not immediately clear as the data is currently presented, furthermore this expression has been correlated with proliferation markers, whereas markers of hypoxia score would be more informative for the proposed mechanism.

The interaction studies are convincing and well performed. One particular strength of the manuscript is the use of enzymatic dead mutants as further negative controls. Furthermore, each interaction is interrogated in depth with exogenous mapping, direct binding assays and endogenous IPs. Some aspects of the manuscript lack detail, particularly the figure legends making interpretation of data difficult at times and would certainly aid readers looking to replicate this data if these are more detailed. Whilst the ubiquitination of DCB1 and subsequent deubiquitination by OTUD5 is clear, the hypothesis that hypoxia promotes SIAH2 binding to DBC1 instead of OTUD5 is somewhat addressed however lacks detail. Hypoxic activation of SIAH2 is not addressed (reported to be via p38 phosphorylation), nor is further detail of OTUD5 binding prior to SIAH2 ubiquitination as the graphical abstract suggests.

The role of this axis in patients and more physiological tumour models is yet to be tested in detail and therefore is a potential weakness of the work proposed. It is currently unclear how important this post-translational regulation of DBC1 is in tumours compared to epigenetic silencing, copy number alterations or other methods of DBC1 silencing. Despite these points, the identification of this axis and detailed characterization provides an important step in further understanding the regulation of this tumour suppressor.

1) The authors propose a hypothesis that OTUD5 is bound to DBC1 in normoxic conditions which is displaced upon activation of SIAH2. This is an interesting hypothesis but requires further experiments to fully support this.

a. Does OTUD5 bind to non-ubiquitinated DBC1? If so, this supports the hypothesis that it is bound in normoxia prior to SIAH2-mediated ubiquitination upon activation in hypoxia as is suggested in the schematic. Direct binding of recombinant DBC1 and OTUD5 could confirm this.

b. Does expression of SIAH2 or OTUD5 change in hypoxia?

c. SIAH2 is still binding in normoxia (Figure 4I), are there other factors still mediating this?

2) It is unclear how prevalent this mechanism is in breast cancer patients. It is difficult to determine how significant this is as is currently presented

a. It may be easier to observe this correlation by scoring staining by H-score and performing correlation from XandY scatter.

b. The authors have correlated staining of xenografts and TMAs with Ki67 positivity and shown clear differences in proliferation. Staining a hypoxic marker (such as CAIX) and observing correlations between DBC1 expression would be a way of validating this mechanism in vivo or in patients.

c. If the main point of regulation in DBC1 expression occurs at the post-translational stage then one would posit there is poor correlation between mRNA and protein levels for DBC1. Perhaps the authors could test this using CPTAC/TCGA data for breast cancer patients readily available on cbioportal.

3) Most experiments are performed in HEK293T, HeLa or MB231 cells. The CHX chase experiments with endogenous DBC1 and endogenous IPs should be repeated with an additional breast cancer line.

4) There is a significant lack of detail in the figure legends throughout. Length of hypoxic exposures are not stated, and it is unclear which cell line is used in Figure 1D. Furthermore, number of replicates and specific statistical tests would be useful in interpreting the data.

5) It would be great to see more detail how OTUD5 was identified as the DUB that deubiquitinates DBC1. Currently the text mentions 'Lots of interaction studies showed' and Figure S2 shows the overexpression of a panel of DUBs tested however there are no further details.

6) In Figure 5A the KO of SIAH2 reduces DBC1 protein levels compared to WT. One would expect that the levels would be equivalent or even slightly increased, could this authors comment on this please?

7) The phenotypic assays in 5B-I were performed in hypoxia. Were these assays also performed in 21% O2? If this data shows no difference, then this strengthens the necessity for this hypoxia dependent axis in the pro-tumorigenic effects. If there are differences observed, then perhaps alternative pathways regulated by SIAH2 or DBC1 are important for tumorigenesis.

8) Can the authors provide more details on Figure 6J? The legend states that patients were stratified by 'DBC1 high expression'. If this is mRNA expression then could the authors explain how this impacts the significance of DBC1 post-translational regulation in breast cancer?

---

## [Author Response]

[Editors’ note: the authors resubmitted a revised version of the paper for consideration. What follows is the authors’ response to the first round of review.]

Unfortunately, numerous significant concerns were raised by each of the 3 reviewers. In considering the sum total of their comments, we are sorry to say that, after consultation with the reviewers, we have decided that this work will not be considered further for publication by eLife. The full details are provided below, but the reviewers raised significant concerns about the exact characterization of DBC1 signaling in breast cancer, the underlying mechanism by which hypoxia regulates this signaling axis and the physiological relevance of findings in breast cancer patients. Without a higher level of enthusiasm, we do not feel this paper would fare well during additional rounds of review at eLife. We hope these comments are useful to you as you consider your next steps for this manuscript.Reviewer #1 (Recommendations for the authors):This study, in general, provides the compelling biochemistry and functional data on examining how SIAH2/OTUD5 may regulate DBC1 protein stability in breast cancer under hypoxia. As a result of DBC1 downregulation of hypoxia, breast cancer may display more malignant phenotype. The study is well designed. However, there are a few things that need to be addressed as suggested below:1. The functional role of DBC1 in breast cancer needs to be characterized in details. For example, the expression of DBC1 in different subtypes of breast cancer. The phenotype of DBC1 overexpression and KO in breast cancer needs to be characterized in details.

We thank the reviewer for positive comments and constructive suggestions. In our manuscript, we have checked the expression of DBC1 in different cell lines including breast cancer, colon cancer, hepatocarcinoma etc. The results showed that hypoxia could induced the degradation of DBC1 in many cell lines (Figure 1F).

In addition, tissue microarray analysis using antibodies against SIAH2 or DBC1 revealed significant negative correlations between the protein levels of DBC1 and SIAH2 in human breast cancer patients (Figure 6E-F), which further validated our results from the bioinformatics analysis. Furthermore, the DBC1 protein levels were negatively correlated with that of Ki67, a marker of cancer cell proliferation, and clinical stages (new Supplementary Table 2), indicating that the quality control of DBC1 under hypoxia play an important role in the breast tumor progression.

2. Throughout the study, two main cell lines are used Hela and MDA-MB-231. Hela cells may display diminished p53 protein due to HPV expression while MDA-MB-231 cells contain mutant p53. It is unclear whether Hypoxia, by downregulating DBC1 level, would lead to further decrease of p53 activity in these cells. This would need to be clarified and reconciled. In addition, some other breast cancer cell lines with WT p53 (at least from in vitro) would need to be added to further strengthen the conclusion.

R2. We agree with the reviewer’s comments. Several reports have shown that DBC1 acts as a native inhibitor of SIRT1 in human cells, and the repression of SIRT1 increases p53 acetylation leading to upregulated p53 function. We analyzed the levels of p53 acetylation and SIAH2-DBC1 signaling in five breast cell lines that have different P53 conditions (MCF10A and MCF7 cells with WT p53, MDA-MB-231, T47D and JIMT1 cells with mutant p53). The results showed that, regardless WT or mutant p53, the acetylation of p53 was decrease after 24-hour hypoxia, which was blocked by SIAH2 knockdown (new Figure 5A).

3. The molecular mechanism by which hypoxia regulates SIAH2/OTUD5 interaction with DBC1 needs more detailed analysis and discussion. Is HIF involved in this process?

R3. Many thanks for the reviewer’s comments. As shown in the new Figure 4I, we found that the deubiquitinase OTUD5 only interacted with DBC1 under normoxic conditions, and the interaction between SIAH2 and DBC1 was increased under hypoxia, which promoted the ubiquitination and degradation of DBC1. We further analyzed the interaction of OTUD5 with DBC1 in *SIAH2* knockout cells, and showed that deletion of SIAH2 significantly increased the interaction of OTUD5 with DBC1 under hypoxic conditions (new Figure 4I).

We have also analyzed whether DBC1 protein levels were affected by a HIF1α inhibitor (CAS 934593-90-5-Calbiochem) in MDA-MB-231 cells under hypoxia. The results showed that hypoxia-induced decrease of DBC1 protein and p53 acetylation were not blocked by HIF1α inhibitor (Figure 1-figure supplement 1A), and suggested that HIF1α signaling pathway might not be involved in DBC1 stability regulation.

Reviewer #2 (Recommendations for the authors):This study reports on the control of DCB1 by the ubiquitin ligase Siah2 under hypoxia growth conditions in breast cancer, while the deubiquitinating enzyme OTUD5 limits DCB1 ubiquitination and thereby prolongs its stability under normoxia.The following are some of the key points raised for authors considerations:It is unclear what underlies the control of DCB1 stability under hypoxia (and not under other conditions).

R4. We thank the reviewer’s comments and suggestions. We have demonstrated that under hypoxic conditions, DBC1 was ubiquitinated by the E3 ubiquitin ligase SIAH2 and degraded via the ubiquitin-proteasome pathway under hypoxia. In this study, we found that hypoxia could promote the interaction between SIAH2 and DBC1 (new Figure 4I). Furthermore, several studies have reported that the ubiquitin ligase activity and stability of SIAH2 were increased under hypoxic conditions, which is due to its phosphorylation mediated by several kinases including p38. Under hypoxic conditions, we found that hypoxia induced DBC1 degradation was partially blocked by Doramapimod, an inhibitor of p38 (new Figure 3-figure supplement 1I). Moreover, we found that inhibition of p38 partially decreased SIAH2 protein and inhibited the interaction between SIAH2 and DBC1 under hypoxia (new Figure 3-figure supplement 1J). Therefore, there results show that hypoxia activates SIAH2 to interact with DBC1, which induces DBC1 degradation by ubiquitin-proteasome system to regulate DBC1 stability.

It would be important to determine what is the physiological significance of DCB1 loss under hypoxia for breast cancer cells.

R5. We have shown that, under hypoxic conditions, the degradation of DBC1 could promote tumor cell proliferation and migration, and inhibit DNA damage-induced apoptosis (new Figures 5B-E). Consistent with this observation, we found that the protein levels of DBC1 in clinical breast cancer samples were closely related with the clinical stage and pathological grades (new Figures 6H, 6I and Supplementary Table 2). These results demonstrate that SIAH2-mediated loss of DBC1 under hypoxia plays an important role in the breast tumor progression.

This work focuses on breast cancer, and it is necessary to better establish relevance in numerous breast cancer (all types?) as opposed to other tumor types.

R6. As following the reviewer’s suggestion, we studied more cancer cell lines, including breast cancer cells (T47D, JIMT1, MCF7, MDA-MB-231, etc) and other type of tumor cells (HepG2, A549, etc). These results show that the degradation of DBC1 protein under hypoxic conditions is a common phenomenon in tumor cells (Figure 1F and new Figure 5A).

The authors claim for dynamic regulation of DCB1 stability, but it is unclear what determines the dynamics.

R7. Thanks for the reviewer’s suggestion. According to the result that SIAH2mediated DBC1 ubiquitination was reversible when restoring the oxygen concentration (Figure 4A). Combined with mapping and interaction results, we found that under hypoxia, SIAH2 replaced OTUD5 for binding to the 1-130 domain of DBC1 (Figures 2F, 4H, 4J and new Figure 4I), which mediated the ubiquitination and degradation of DBC1 protein. These results suggest that hypoxia induces DBC1 degradation via the ubiquitin-proteasome system was mediated by E3 ligase SIAH2. We agree with the reviewer that the word “dynamic” may be confusing and have revised the title to “hypoxia induces DBC1 proteasomal degradation by SIAH2 in breast cancer progression”.

Experiments are performed largely under one condition and not another; complementing the assessment for hypoxia with normoxia conditions and vice versa is needed for number of studies shown.

We are agreeing with the reviewer. Actually, these experiments in our manuscript were strictly carried out under normoxic and hypoxic conditions. Now, we included the normoxia data set (new Figures 5B-E and Figure 5-figure supplement 1).

In this study, Liu et al. report on post-translational regulation of DBC1 by the ubiquitin ligase Siah2 and the DUB OTUD5, which occurs during hypoxia. The separate roles of Siah2 and DBC1 in tumor progression have been shown in previous publications. Here the authors define the role of Siah2 in regulating DBC1 stability in breast cancer cells during hypoxia. While interesting, several points need to be experimentally addressed before further evaluation of this study as a candidate for publication in eLife should take place.– Why is Siah2 more effective in degrading DBC1 under hypoxia? The title and elsewhere in this manuscript authors claims for dynamic regulation of DBC1 stability. What underlies this dynamic regulation?

R9. It has been reported that under hypoxic conditions, the regulation of SIAH2 was regulated by several mechanisms including gene transcription, post transcriptional modifications, and formation of dimeric complexes. SIAH2 could be phosphorylated by kinases such as p38 and Src under hypoxic conditions, increasing its ubiquitin ligase activity and protein stability^[1, 2]^. Combined with Co-IP, ubiquitination and degradation assays, we found that hypoxia activated SIAH2 to interact with DBC1 and promote its degradation by the ubiquitin-proteasome pathway (Figure 3H and new Figure 4I). We agree with the reviewer that the word “dynamic” may be confusing and have revised the title to “hypoxia induces DBC1 proteasomal degradation by SIAH2 in breast cancer progression”.

– The work suggests that the Siah2 effect on DBC1 occurs in breast cancer cells, but most of the data is based on studies performed in either 293 or HeLa cells. Work on breast cancer lines is primarily limited to a single cell line.

R10. We are agreeing with reviewer’s comments. Besides using 293T or HeLa cells, we have also performed experiments in normal and breast cancer cell lines (MCF10A, MCF7 and MDA-MB-231), including the Co-IP assay (Figures 2D, 4B and Figure 2-figure supplement 1B, Figure 2-figure supplement 1C, Figure 3-figure supplement 1J and Figure 4-figure supplement 1F), detecting the ubiquitination level of DBC1 (Figures 3F and Figure 4A) and the protein level of DBC1 (Figures 1C-F, Figures 3H-L, and Figure 1-figure supplement 1A-C, Figure 3-figure supplement 1F-I and Figure 5A). In addition, we analysis DBC1 protein expression levels and DBC1 downstream signaling pathway activities in various breast cancer tumor cell lines under hypoxic conditions (Figure 1F and new Figure 5A). These results showed that SIAH2-mediated degradation of DBC1 protein under hypoxic conditions generally occurred in many breast cancer cells.

– Are the effects reported here also seen in breast cancer patients?

R11. Following this suggestions, we analysis the protein expression database and tissue microarray samples of human breast cancer patients, and found that the protein expression levels of SIAH2 and DBC1 were significantly negatively correlated (Figure 6E and Figure 6-figure supplement 1A), and the protein level of DBC1 was significantly correlated with the clinical stage and pathological grade (new Figures 6H, 6I and new Supplementary Table 2). It is demonstrated that SIAH2-mediated ubiquitination and degradation of DBC1 plays an important role related in breast tumor progression in human.

– Much of the data relies on overexpression. Authors ought to demonstrate changes in endogenous proteins and their physiological relevance (when does it happen and why).

R12. We thank the reviewer’s suggestion. In our manuscript, the ubiquitination level and protein stability of endogenous DBC1 were detected in both MCF7 cells and MDAMB-231 cells (Figures 3F and 4A, Figures 1C-F, Figures 3H-L, and Figure 1-figure supplement 1A-C, Figure 3-figure supplement 1F-I and Figure5A). Also, the endogenous interaction between SIAH2, DBC1 and OTUD5 were performed in MCF10A, MCF7 and MDA 231 cells (Figure 2D, Figures 4B and Figure 2-figure supplement 1B, C and Figure 4-figure supplement 1F). these results show that under normoxic conditions, endogenous protein of DBC1 binds to the deubiquitinase OTUD5 and maintains a non-ubiquitinated stable state. Under hypoxic conditions, DBC1 binds to the E3 ubiquitin ligase SIAH2, increases the level of ubiquitination modification, and is further degraded by the proteasome pathway. The detailed mechanisms underlying hypoxia-mediated OTUD5 and SIAH2 regulation are under active investigation in our lab.

– OTUD5 is shown to regulate DBC1 under normoxic (and not hypoxic?) conditions. Why?

R13. We showed that OTUD5 binds to and deubiquitinates DBC1 under normoxia, leading to enhanced protein stability of DBC1. However, such OTUD5-DBC1 interaction is attenuated under hypoxia by multiple mechanisms. First, SIAH2 could attenuate endogenous OTUD5-DBC1 interaction under hypoxic conditions, while deletion of SIAH2 significantly increased the interaction of OTUD5 with DBC1 under hypoxia (new Figure 4I). Therefore, these results suggest that OTUD5 is responsible for DBC1 deubiquitination and stabilization under normoxic conditions.

– All figure legends and Results sections lack details. It is not clear how the experiments were carried out.

R14. Thanks for the suggestions, we have carefully checked all of the legends and methods, and revised them with experimental technical details and experimental conditions. New and updated text is highlighted in red.

Additional comments:Figure 1: the authors assess the overlap of differentially expressed genes between hypoxia and DBC1 deletion (in normoxia). DEG needs to be compared under the same (hypoxic) conditions. Can overexpression of DBC1 rescue phenotypes be seen under hypoxia?

R15. Thanks for reviewer’s suggestion. In order to identify the candidate genes that are both regulated by hypoxia and DBC1, we analyzed the DEGs of wild-type cells under normoxic and hypoxic conditions, and the DEGs of wild-type and DBC1 knockout cells under normoxic conditions, respectively. (Figures 1H and 1I). Since hypoxia also leads to DBC1 loss in WT cells, we will not be able to separate hypoxiaregulated genes from that regulated by DBC1 loss under hypoxia condition.

Figure 2A: the authors overexpress several E3 ubiquitin ligases to determine possible candidates responsible for DBC1 protein degradation. However, the level of expression of each E3 ligase (mRNA level) is missing, making it difficult to compare the efficiency of each E3 ligase in degrading DBC1. HUWE1 also seems to be as efficient as Siah2 in degrading DBC1, but it is not mentioned in the Results section. Why? The experiment is carried out in normoxia rather than hypoxia; why? Does any UBL tested exhibit different efficiency for DBC1 interaction and degradation under hypoxia?

R16. Thanks for suggestions. To screen the ubiquitin ligases of DBC1, we overexpressed a series of hypoxia-related ubiquitin ligases and found that both HUWE1 and SIAH2 might mediated DBC1 stability regulation (Figure 2A). In the revised Figure 2-figure supplement 1A, we have included Western blot results to show the protein levels of all E3s overexpressed. We agree with the reviewer that normoxia is not the perfect condition for E3 screening assay. Therefore, we performed the same assay under hypoxia and found that only SIAH2 was able to reduce DBC1 protein levels (new Figure 2B). These data clearly demonstrate that SIAH2 is the E3 ligase targeting DBC1 under hypoxia.

Figure 3: The authors should demonstrate, experimentally, why Siah2 dependent degradation of DBC1 is primarily claimed to be under hypoxia.

R17. We thank the reviewer’s suggestion. Please refer to the detail molecular mechanism of SIAH2 dependent degradation of DBC1 in R6 and R11.

Figure 3H: the authors compared DBC1 protein levels in WT and Siah2 KO cells using MDA-MB-231 cells. It would be informative to compare the level of DBC1 between WT and Siah2 KO in the same Western and demonstrate possible changes in the level of DBC1 under normoxia. Additional breast cancer cell lines (and non-transformed breast cultures) should be studied.Figure 3K: It would be informative to perform CHX chase under normoxia, compared with CHX done under hypoxia. These assessments need to be extended to include additional breast cancer cell lines.Figure 4: the authors should study OTUD5 binding to DBC1 under hypoxia compared with normoxia to substantiate the claim for preferred interaction under normoxia. Does OTUD5 expression change in breast cancer patients, compared to healthy donors or between cancerous and normal tissues? Does it correlate with patient survival? The screening of DUBS to identify the candidate responsible for deubiquitination of DBC1 is not clear. Sup Figure 4 A shows the level of expression of the different DUBS but does not reveal why the authors chose to focus on OTUD5.

R18. Thanks for good suggestions. We analyzed the interaction of endogenous OTUD5 and DBC1 under normoxia and hypoxia. The result showed that OTUD5 only interacted with DBC1 under normoxia in WT cells. However, such interaction in SIAH2-KO cells also took place under hypoxia (new Figure 4I). OTUD5 was also degraded via the ubiquitin-proteasome pathway under hypoxia. These results suggest that, under hypoxic conditions, the interaction between SIAH2 and DBC1 attenuates OTUD5 binding to DBC1, and the degradation of OTUD5 further reduces its interaction with DBC1.

Unfortunately, there is no good immunohistochemical antibodies for OTUD5 to check its expression levels in clinical samples (we have check the OTUD5 antibodies from Proteintech: 21002-1-AP, CST: OTUD5 (D8Y2U) Rabbit mAb #20087 and Abcam: ab254742 as same as the references, but those not work on breast cancer tissue). Data analysis using TCGA and CPTAC database revealed that the mRNA level of OTUD5 was not obvious changed (Figure 6B). We have included the detail of the DUBs screening. In order to screen and identify the DUB of DBC1, we constructed human DUBs plasmids and detected their overexpression level in cells. The interaction assays between DUBs and DBC1 were performed and found that OTUD5 and USP29 are potential deubiquitinases of DBC1 (Figure 4-figure supplement 1A and B). Further deubiquitination experiments results showed that only OTUD5 could specifically deubiquitinate DBC1 (Figure 4-figure supplement 1C). So, we identify OTUD5 as the deubiquitinating enzyme of DBC1.

Figure4I: the authors should add Siah2 KO cells to their analysis; it is important to determine whether OTUD5 and Siah2 compete for binding to DBC1.

R19. We thank the reviewer for good suggestion. We performed the study and found that deletion of SIAH2 significantly increased the interaction of OTUD5 with DBC1 under hypoxia (new Figure 4I). These results suggest that under hypoxic conditions the interaction between SIAH2 and DBC1 attenuates OTUD5 binding to DBC1.

Figure 5B, D, F: the authors should address whether Siah2 KO can promote apoptosis, colony formation assay, and invasion under normoxia, not only in hypoxia.

We are agreeing the reviewer, and have included new data for the control groups under normoxia and hypoxia (new Figures 5B-E and Figure 5-figure supplement 1).

Figure 5M: the authors should check for possible changes in apoptosis, in addition to shown data for proliferation.

R21. We have updated the results of apoptosis by detecting TUNEL staining on breast cancer tissue (new Figure 5I).

All figure legends and Results sections lack details. It is not clear how the experiments were carried out. (for example, Figure 1A: how long were the cells kept in hypoxia when RNAseq was performed? The authors should define the conditions used to reach hypoxia in the results and Figure legends. Condition of 1% O2 is briefly noted in the Methods section, but not in results or legends.

R22. Thanks for suggestions, we have carefully checked all of the legends and methods, and updated experimental technical details and experimental conditions.

Reviewer #3 (Recommendations for the authors):The authors sought to identify how the breast cancer tumour suppressor DBC1 (CCAR2) is post-translationally regulated. This came from transcriptomic analysis of MB231 cells in hypoxia revealed decreased SIRT1 activity, without changes in protein levels. Instead known regulator DBC1 displayed reduced protein expression in hypoxia with no changes in mRNA. Through targeted screen of hypoxia associated E3 ligases and IP-MS of SIAH2, the authors identify interaction between DBC1 and SIAH2. They comprehensively map the interaction between DBC1 and SIAH2 and identify the residue ubiquitinated. Upon 6h of reoxygenation the ubiquitination of DBC1 is decreased. OTUD5 was identified as the DUB that deubiquitinates DBC1. The authors propose that in normoxia, OTUD5 is bound to DBC1 which is displaced by SIAH2 upon activation by hypoxia, however despite the comprehensive study of the interaction between DBC1 this is not fully supported by the data. SIAH2 Knockout reduced colony formation, cell migration and growth in a xenograft tumour model which could all be partially rescued by knockout of DBC1. SIAH2 expression was elevated in breast cancer and correlated with cancer stage. Through staining of a TMA the authors suggest a negative correlation between DBC1 and SIAH2 protein expression however this is not immediately clear as the data is currently presented, furthermore this expression has been correlated with proliferation markers, whereas markers of hypoxia score would be more informative for the proposed mechanism.The interaction studies are convincing and well performed. One particular strength of the manuscript is the use of enzymatic dead mutants as further negative controls. Furthermore, each interaction is interrogated in depth with exogenous mapping, direct binding assays and endogenous IPs. Some aspects of the manuscript lack detail, particularly the figure legends making interpretation of data difficult at times and would certainly aid readers looking to replicate this data if these are more detailed. Whilst the ubiquitination of DCB1 and subsequent deubiquitination by OTUD5 is clear, the hypothesis that hypoxia promotes SIAH2 binding to DBC1 instead of OTUD5 is somewhat addressed however lacks detail. Hypoxic activation of SIAH2 is not addressed (reported to be via p38 phosphorylation), nor is further detail of OTUD5 binding prior to SIAH2 ubiquitination as the graphical abstract suggests.The role of this axis in patients and more physiological tumour models is yet to be tested in detail and therefore is a potential weakness of the work proposed. It is currently unclear how important this post-translational regulation of DBC1 is in tumours compared to epigenetic silencing, copy number alterations or other methods of DBC1 silencing. Despite these points, the identification of this axis and detailed characterization provides an important step in further understanding the regulation of this tumour suppressor.1) The authors propose a hypothesis that OTUD5 is bound to DBC1 in normoxic conditions which is displaced upon activation of SIAH2. This is an interesting hypothesis but requires further experiments to fully support this.a. Does OTUD5 bind to non-ubiquitinated DBC1? If so, this supports the hypothesis that it is bound in normoxia prior to SIAH2-mediated ubiquitination upon activation in hypoxia as is suggested in the schematic. Direct binding of recombinant DBC1 and OTUD5 could confirm this.

R23. Many thanks for the reviewer’s suggestion and we have found OTUD5 could direct bind non-ubiquitinated DBC1. in vitro, pull-down experiments were performed using purified His-DBC1 protein after co-incubation with cell lysates under normoxic conditions (Figure 4C). This result showed that OTUD5 had a direct interaction with DBC1. And as shown in the new Figure 4I, OTUD5 interacted with DBC1 under normoxic conditions. In order to further address this concern, we have performed new Co-IP experiments, and showed OTUD5 could bind to non-ubiquitinated DBC1 (Author response image 1).

**Author response image 1. sa2fig1:** HEK293T cells were transfected with Myc-DBC1 (K287R) mutant and Flag-OTUD5 for 24 h.Cells were collected for immunoprecipitation with anti-Flag antibody.

b. Does expression of SIAH2 or OTUD5 change in hypoxia?

R24. It has been reported that under hypoxic conditions, the regulation of SIAH2 was regulated by several mechanisms including gene transcription, post transcriptional modifications, and formation of dimeric complexes.The detailed mechanisms underlying hypoxiamediated OTUD5 and SIAH2 regulation are under active investigation in our lab.

c. SIAH2 is still binding in normoxia (Figure 4I), are there other factors still mediating this?

R25. In our manuscript, the endogenous interaction assay between SIAH2 and DBC1 was performed in MDA-MB-231 cells supplemented with the proteasomal inhibitor MG132 (10 μM) before cells were harvest. Because MG132 could inhibit SIAH2 degradation and promote its stability, so that there was weak interaction of SIAH2 and DBC1 could be detected under normxia. The direct interaction between SIAH2 and DBC1 was confirmed by pull-down experiments using purified proteins in vitro (Figures 2D, G and H). This has been clarified in the updated figure legend.

2) It is unclear how prevalent this mechanism is in breast cancer patients. It is difficult to determine how significant this is as is currently presenteda. It may be easier to observe this correlation by scoring staining by H-score and performing correlation from XandY scatter.

R26. Many thanks for the suggestion. Tissue microarray analysis using antibodies against the SIAH2 or DBC1 revealed that the expression of DBC1 was significantly negatively correlated with that of SIAH2 in clinical breast cancer samples (Figures 6E, 6F). Furthermore, we found that the expression of DBC1 was negatively correlated with clinical stage and the percentage of the Ki67-positive cell population (new Figures 6H, 6I and new Supplementary Table 2).

b. The authors have correlated staining of xenografts and TMAs with Ki67 positivity and shown clear differences in proliferation. Staining a hypoxic marker (such as CAIX) and observing correlations between DBC1 expression would be a way of validating this mechanism in vivo or in patients.

R27. These are very helpful suggestions. We have performed immunofluorescence assay using a CAIX monoclonal antibody (Proteintech, 66243-1-Ig) to analyze DBC1 expression levels in the hypoxic zone in clinical breast cancer samples. The result showed that the DBC1 expression is dramatically reduced in the CAIX^+^ hypoxic zone (new Figure 6G), demonstrating the DBC1 loss due to hypoxia is clinically significant.

c. If the main point of regulation in DBC1 expression occurs at the post-translational stage then one would posit there is poor correlation between mRNA and protein levels for DBC1. Perhaps the authors could test this using CPTAC/TCGA data for breast cancer patients readily available on cbioportal.

R28. We analyzed the correlation between mRNA and protein levels for DBC1 on cbioportal for breast cancer patient samples, generated by CPTAC. Surprisingly, there was a positive correlation between mRNA and protein levels of DBC1. We think there are two possible explanations. (1) As we have shown that DBC1 reduction only takes place in the hypoxic regions in a tumor (new Figure 6G), the DBC1 expression data from cbioportal may not be limited to the hypoxic regions. (2) The tumor microenvironment could have more factors besides hypoxia stress.

3) Most experiments are performed in HEK293T, HeLa or MB231 cells. The CHX chase experiments with endogenous DBC1 and endogenous IPs should be repeated with an additional breast cancer line.

R29. We agree with the reviewer. We have examined the endogenous protein interaction, ubiquitination level and protein level of DBC1 in breast cancer cell lines. The results of CHX chase experiments with endogenous DBC1 and endogenous CoIPs in breast cancer cells were shown in Figures 2D, 3K, 4B and new Figures Figure 2-figure supplement 1B and C, Figure 3-figure supplement 1F-J and 4I.

4) There is a significant lack of detail in the figure legends throughout. Length of hypoxic exposures are not stated, and it is unclear which cell line is used in Figure 1D. Furthermore, number of replicates and specific statistical tests would be useful in interpreting the data.

R30. Thanks for suggestions, we have carefully checked all of the legends and methods, and updated experimental technical details, experimental conditions and detail protocol information.

5) It would be great to see more detail how OTUD5 was identified as the DUB that deubiquitinates DBC1. Currently the text mentions 'Lots of interaction studies showed' and Figure S2 shows the overexpression of a panel of DUBs tested however there are no further details.

R31. Thanks for suggestions. We have included the detail of the DUBs screening. In order to screen and identify the DUB of DBC1, we constructed human DUBs plasmids and detected their overexpression level in cells. The interaction assays between DUBs and DBC1 were performed and found that OTUD5 and USP29 are potential deubiquitinases of DBC1 (new Figures S4A and S4B). Further deubiquitination experiments results showed that only OTUD5 could specifically deubiquitinate DBC1 (new Figure S4C). So, we identify OTUD5 as the deubiquitinating enzyme of DBC1.

6) In Figure 5A the KO of SIAH2 reduces DBC1 protein levels compared to WT. One would expect that the levels would be equivalent or even slightly increased, could this authors comment on this please?

In order to address this concern and confirm the effect of SIAH2 deletion on DBC1 protein level, we detected the levels of DBC1 under hypoxia in several breast cancer cell lines with or without siRNA of *SIAH2*, such as MCF10A, MCF7, T47D and JIMT1 cells (new Figure 5A). The results show that DBC1 was degraded under hypoxic conditions in breast cancer cells, which was dependent on SIAH2.

7) The phenotypic assays in 5B-I were performed in hypoxia. Were these assays also performed in 21% O2? If this data shows no difference, then this strengthens the necessity for this hypoxia dependent axis in the pro-tumorigenic effects. If there are differences observed, then perhaps alternative pathways regulated by SIAH2 or DBC1 are important for tumorigenesis.

Thanks, we agree with the reviewer. We included normoxia data as shown in revised Figure S5. Under normoxic conditions (21% O_2_), *SIAH2* knock out did not affect tumor cell proliferation and DNA damage-induced apoptosis. However, the deletion of SIAH2 and DBC1 significantly increased cell proliferation and reduced Etoposide-mediated apoptosis (new Figures 5B-E and Figure S5).

8) Can the authors provide more details on Figure 6J? The legend states that patients were stratified by 'DBC1 high expression'. If this is mRNA expression then could the authors explain how this impacts the significance of DBC1 post-translational regulation in breast cancer?

R34. Thanks for the suggestion. The “High or low levels” represents mRNA expression of DBC1. As we have shown that DBC1 reduction only takes place in the hypoxic regions in a tumor (new Figure 6G) and the total mRNA levels of DBC1 might not accurately characterize protein levels, so we removed these data.

References

1. Sarkar, T.R., et al., Identification of a Src tyrosine kinase/SIAH2 E3 ubiquitin ligase pathway that regulates C/EBPdelta expression and contributes to transformation of breast tumor cells. Mol Cell Biol, 2012. 32(2): p. 320-32.

2. Khurana, A., et al., Regulation of the ring finger E3 ligase Siah2 by p38 MAPK. J Biol Chem, 2006. 281(46): p. 35316-26.